# Monolithic electrostatic actuators with independent stiffness modulation

Yuejun Xu [1], Jian Wen[1,2], Etienne Burdet [1] & Majid Taghavi [1,3] ✉

Robotic artificial muscles, inspired by the adaptability of biological muscles, outperform rigid robots in dynamic environments due to their flexibility. However, the intrinsic compliance of the soft actuators restricts force transmission capacity and dynamic response. Biological muscle modulates their stiffness and damping, varying viscoelastic properties and force in interaction with the surroundings. Here we replicate this function in the electro-stiffened ribbon actuator, a monolithic strong actuator capable of high contraction and stiffness modulation. electro-stiffened ribbon actuator employs dielectric-liquid-amplified electrostatic forces for contraction, and electrorheological fluid for rapid (<10 ms) stiffness and damping adjustments. This seamless integration enables contractile force modulation, extending its capability as a lightweight variable resistance passive spring by over 2.5 times, and improves its dynamic responses, with faster contractions and rapid attenuation of oscillations by more than 50%. We demonstrate electro-stiffened ribbon actuator's versatility in active, passive and dual connection functions, including arm-bending wearable robotics, robotic arms with variable impact resistance and muscle-like stiffness and damping modulation.

Inspired by nature, robots made of flexible materials and integrating soft actuators are deformable and inherently compliant, increasing adaptability in interaction with human bodies[1,2] and unstructured environments[3,4]. A distinguishing feature of biological muscles that should equip soft actuators is their ability to modulate the mechanical impedance or viscoelasticity to achieve desired functions[5]. For example, Octopus arms[6] and elephant trunks[7] selectively harden parts of their constituents to realize delicate manipulation and deliver high force; runners vary their leg impedance to stabilize their gait and alter their stride frequency[8]; humans modulate their muscle stiffness to stabilize movement[9] and optimize haptic perception[10].

Previous studies have attempted to combine the benefits of structural compliance with the convenience and versatility of achieving compliance through control. In rigid body robots, this is achieved using variable stiffness actuators[11–13], including active stiffness adjustment using antagonistic actuation schemes[14] and spring mechanisms[12], resulting in faster and safer motion control[15,16]. In inherently compliant soft robots, variable compliance is achieved by several active

reinforcement methods. Pneumatic[17,18] or hydraulic[19] fluid actuators vary their stiffness by pressure changing or being arranged in an agonist-antagonistic pair. In granular or layer jamming, particles or lamina are jammed together to increase stiffness by friction force while being vacuumed[20–23]. Smart materials such as low-melting-point alloys[24] and shape memory alloys[25] tune their stiffness through phase changes driven by temperature. These reinforcement methods enable the soft robot to increase the force delivered to the environment when necessary, which has been demonstrated in various applications spanning manipulators[22,26], wearable devices[27,28], and invasive surgery robots[29]. However, these technologies often suffer from fundamental drawbacks such as slow response times or the need for bulky components, leading to increased design complexity[30].

Electro-responsive materials employed for actuation or stiffening are known for their rapid response times and direct electrical control. Recent advancements in electrostatic actuators have introduced alternative actuation approaches in soft robotics. For instance, HASEL actuators[31] demonstrate the potential for large-scale artificial

[1]Department of Bioengineering, Imperial College London, London, UK. [2]State Key Laboratory of Electrical Insulation and Power Equipment, Xi'an Jiaotong University, Xi'an, China. [3]School of Engineering and Materials Science, Queen Mary University of London, London, UK. ✉e-mail: m.taghavi@imperial.ac.uk

muscles[32], and electro-ribbon actuators (ERAs)[33–35] incorporate liquid-amplified electrostatic forces with elastic origami structures for high-contraction applications. Auxetic electrostatic actuators (AELAs)[36] employs this concept in an auxetic structure to enhance impact damping properties, mimicking human muscle behavior. However, the generated force of these actuators is constrained by their initial material stiffness, for example, ERA and AELA both require replacement of high-stiffness electrodes or elastomers to enhance force output, and often at the cost of reduced displacement. Consequently, while certain electrostatic actuators may excel in specific performance metrics, no single approach has yet replicated the multifunctional capabilities of biological muscles. To address this limitation, a promising approach is incorporating electro-active variable stiffness control into electrostatic actuators, which allow for dynamically adjusting force and displacement capacity while maintaining full electrical control and rapid response time. Electrostatic clutches, which operate at similar voltages, are typically used for stiffness modulation in the tensile direction due to their frictional properties[37]; however, achieving large stiffness variation in other directions, such as bending, typically requires multiple layers[38]. Electrorheological fluids (ERFs) materials[39], are a class of smart materials whose viscoelastic properties can change significantly under an electric field[40–44]. The stiffness variation characteristics of ERFs have garnered research attention for their application in soft structures. Sun et al. used ERF to develop a layer-jamming device with variable stiffness, increasing the stiffness variation capacity by 1.9 times[45]. Jing et al. introduced a variable stiffness soft beam using ERFs, which responds rapidly (under 65 ms) and exhibits substantial stiffness variation (over 1500% under 4.5 kV/mm)[46]. These studies collectively underscore the effective stiffness modulation achievable with ERF.

This study introduces an innovative method for achieving controllable variable stiffness in electrostatic actuators by leveraging electrorheological fluid. Referred to as the Electro-Stiffened Ribbon Actuator (ESRA), this actuator operates through electrostatic force, simultaneously adjusting the bending stiffness of beam electrodes through encapsulated ERF. This dual mechanism of electrostatic actuation and stiffness modulation via ERF encapsulation in a monolithic structure marks a significant advancement in soft robotic actuator design, outperforming advanced bionic muscles in several aspects: (i) ESRA's variable stiffness enables continuous force adjustments and enhances dynamic response, while its ability to function as a passive element allows for large deformations and tunable tensile elasticity. (ii) By increasing the stiffness voltage, ESRA can quickly suppress impacts and disturbances, effectively minimizing prolonged oscillations. (iii) Configured in agonist-antagonist pairs or series, ESRA's controllable damping and stiffness characteristics significantly enhance its adaptability.

## Results

### Working principle of ESRA

Biological muscles act as unidirectional motors during shortening and as controllable variable resistances during lengthening[47]. This behavior originates from the cross-bridges linking structure within muscle fibers: (1) Filament overlap decreases at long lengths, resulting in a reduction in active force; (2) Connective tissues are stretched at longer lengths, resulting in increased passive force[48]. This muscle behavior can be mathematically described represented as[48]:

$$\mu = K(u)\left[\lambda_r(u) - \lambda\right] - D(u)\dot{\lambda}, \ \lambda_r(u) = \lambda_r - au, a > 0 \tag{1}$$

where $\mu$ is muscle tension; $K$ and $D$ represent the muscle's stiffness and viscosity, respectively, which increase with activation, and $\lambda$ the muscle length, which tends to shorten with activation. ERA demonstrates force-length characteristics[49] similar to those of biological muscles,

while ESRA allows for modulation of stiffness and damping properties through the use of electro-rheological fluid.

The structure of the ESRA is shown in Fig. 1a. It is actuated by the dielectrophoretic liquid zipping concept[35], which harnesses amplified electrostatic force to close the gap between two electrically insulated electrodes, like a zipper. When oppositely charged, these electrodes create a strong electric field[50], inducing electrostatic attraction that draws them at the initial joint, or zipping point, gradually bringing them together until fully closed. At these zipping points, small droplets of liquid dielectrics, such as silicone oil, is placed to significantly amplify the electrostatic force. These liquid dielectrics possess higher permittivity and breakdown strength compared to air, enabling stronger and sustained electric fields, thereby enhancing the zipping force.

In conventional ERA, there is a trade-off between stroke and load: high-stiffness electrodes can provide high contractile force with a smaller stroke; the stiffness electrode can produce a large stroke but with limited force[35]. This restricts the actuators' ability to accommodate scenarios where there is a broader variation in force and displacement. In contrast, ESRA uses the electrorheological fluid to provide stiffness adjustment capability. These fluids are colloidal suspensions composed of electrically polarizable particles suspended in a dielectric liquid medium. When an electric field is applied, the suspended particles become polarized, experiencing attractive forces between particles of opposite polarity. This leads to the rapid (<10 ms) formation of particle chains[43] along the electric field lines (Fig. 1a), effectively making it behave more like a solid on a macro scale[51,52].

ERF enables stiffness adjustments of the ribbons by applying a stiffening voltage of a similar magnitude (~ kV) to the actuation voltage of ERAs[35,43]. As illustrated in Fig. 1b, when a voltage is applied to the electrodes on two sides of the ERF layer, the ERF material becomes stiffer; when a voltage is applied to the middle two electrodes, the generated electric field between these electrodes induces an electrostatic attraction that initiates the actuator's 'zipping' behavior. The actuator's stiffness can be swiftly and independently tuned or simultaneously actuated and stiffened, through precise voltage control across these four electrodes. This dual-mode functionality enhances the actuator's versatility, enabling the ESRA to operate in two distinct modes: as an active actuator producing contractile force through electrostatic zipping, and as a nonlinear spring providing electrically adjustable passive resistance against external forces.

### Modelling

To investigate the impact of stiffness variation on actuation performance, we developed and validated a model that predicts quasi-static contractile force as a function of extension. Rather than relying on a standard capacitor model to calculate electrostatic attraction, we develop an electrical model that accounts for the time-dependent variations of the electric field through the dielectric relaxation properties of dielectric materials[53]. The mechanical beam model builds upon previous frameworks established for electro-ribbon actuators[49]. These electrical and mechanical models were then coupled to create an integrated predictive framework. Model validation was performed using three commonly employed dielectric materials in electrostatic actuators (see Supplementary Table 3). This model categorizes the actuator's motion into two distinct modes (Fig. 2a): passive and active. In the passive mode, the actuator's contraction is driven solely by the mechanical elastic force, which arises from the beam's nonlinear deflection. In the active mode, the contraction results from a combination of this elastic force and the electrostatic force. The electrostatic force is expressed as

$$F_e = \frac{1}{2} w \varepsilon_0 \varepsilon_{di} E_{di}{}^2 \tag{2}$$

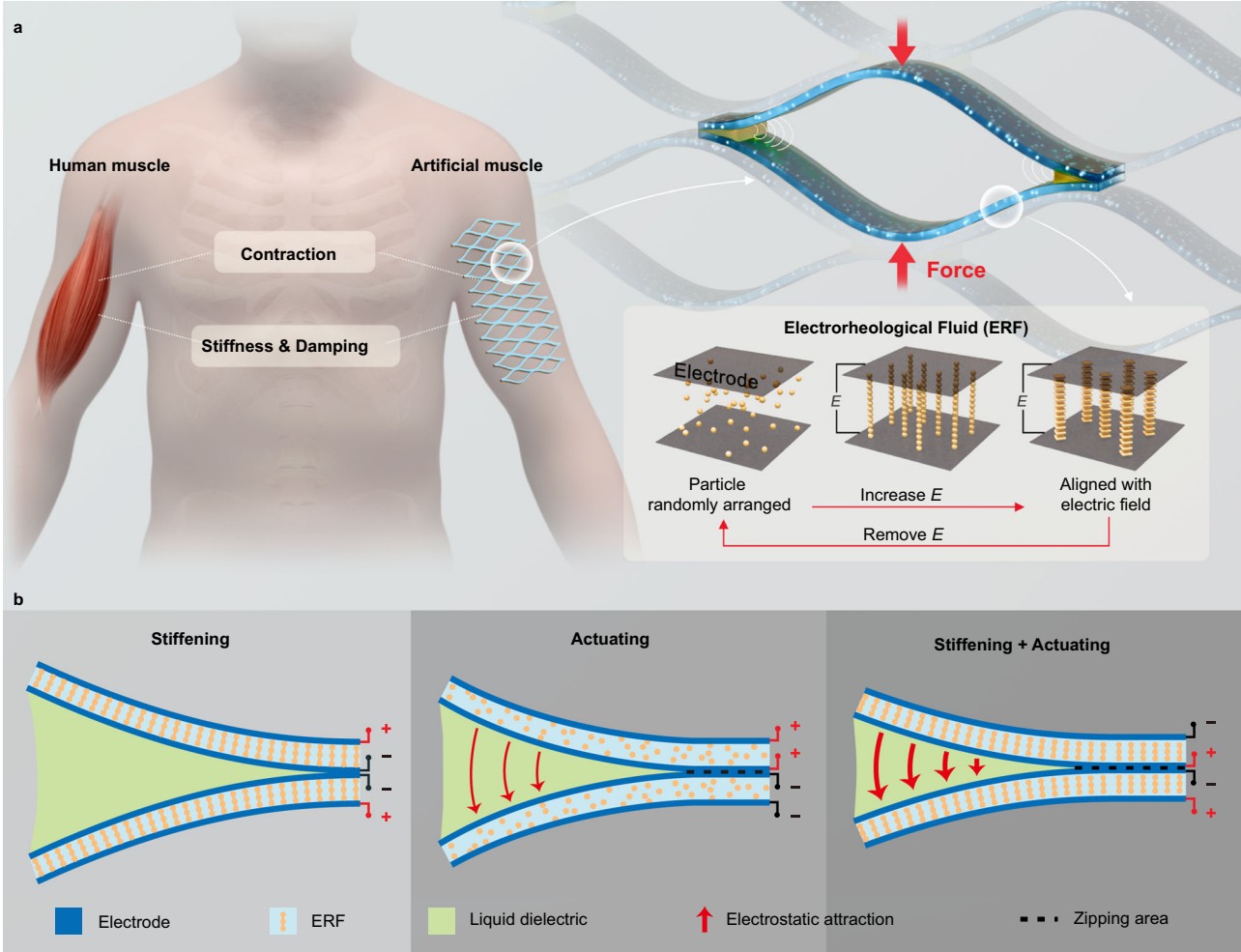

**Fig. 1 | Design and operational principle of the Electro-Stiffened Ribbon Actuator (ESRA). a** Schematic illustrating the structure of the ESRA and the working mechanism of electrorheological fluid (ERF) within the electric field ($E$). **b** Operational principles of ESRA at different states: stiffening, actuation, and combined stiffening and actuation.

where $\varepsilon_{di}$ is the relative permittivity of liquid dielectric, and $\varepsilon_0$ is the permittivity of free space. $w$ is the width of electrodes and $E_{di}$ is the electric field generated in the liquid dielectric under applied voltage. The charge relaxation effect causes the electric field to change dynamically over time, generating a significant electrostatic force between our insulating material. The detailed model and analysis are provided in Supplementary Information (Supplementary Figs. 8–12).

The electrode's bending stiffness modulates the force at both passive and active states. As shown in Fig. 2a, in the passive state, where the actuator functions like a spring, a higher $EI/w$ value denotes a more significant elastic force. Consequently, the contractile elastic force quadruples when the $EI/w$ value increases from 0.002 Nm to 0.008 Nm. In the active state, the stiffer actuator delivers an enhanced actuation force. When a voltage of 10 kV is applied, the force in the active state amplifies to six times that of the passive state, and further stiffness increases, leading to a 1.5-fold enhancement in force.

Based on sandwich beam theory (Supplementary Fig. 9), we determine the increasing effective bending stiffness of the ERF beam when increasing the stiffening voltage, as shown in Fig. 2b (see the mathematical model in the Supplementary Information). In comparison, Fig. 2d displays the actual deflection of an ERF beam, which reduces from 12 mm at 0 kV to 4 mm at 8 kV under identical conditions to the simulation. The measured stiffness values (Fig. 2b) are consistently lower than theoretical predictions across all voltages, likely due to manufacturing imperfections and the increasing nonlinearity of

large deformation. We then employ the developed electromechanical model to simulate the contractile force generated by the actuator under different equivalent bending stiffness (Supplementary Fig. 10). Figure 2c shows the modeled force-extension characteristics of the actuator with various normalized effective bending stiffness. The actuator exhibits a spring-like response as indicated by the passive force, which increases with extension. The graph shows a convergence point between the solid and dashed lines at a certain extension. Beyond this point, the lines merge, indicating that the passive force dominates at higher extensions, thereby preventing contraction under 6 kV voltage. Conversely, reductions in extension prior to this convergence lead to an increase in contraction force. This suggests that the actuator can fully contract under a steady external load before it reaches a certain extension. However, beyond this critical extension point, contraction becomes impractical, making this extension the maximum actuation extension. A higher bending stiffness not only denotes a more significant passive elastic force but also results in a greater actuation force due to a larger $F_e$ value at the zipping point. This is a consequence of a reduced zipping angle and $d_{di}$. Changes in the beam's stiffness dynamically shift the contraction force among various curves.

## Characterization of the ERF beam
Figure 3 shows the variable stiffness characteristics of ERF beams under different stiffening voltages with varying loads. In contrast to

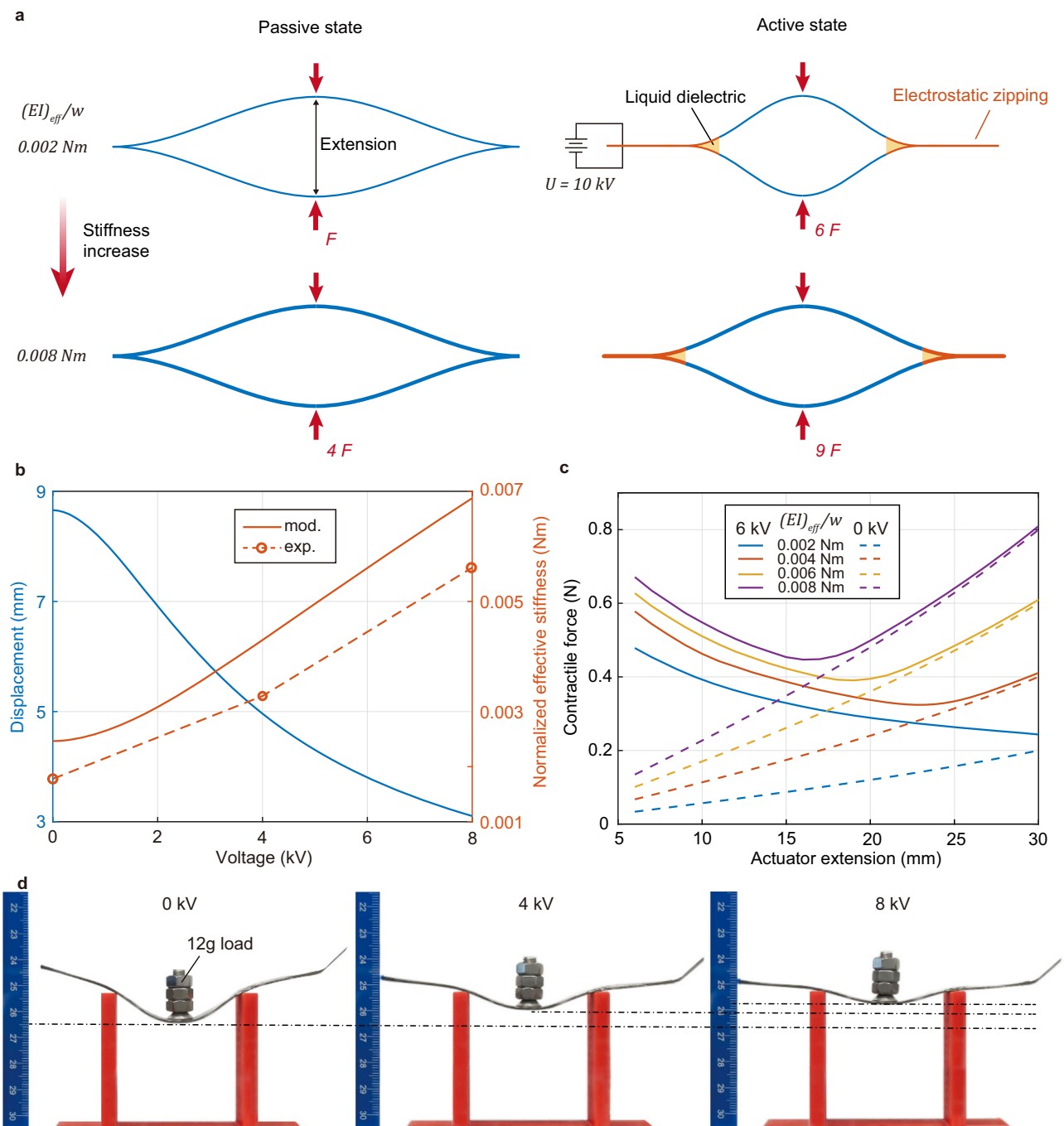

**Fig. 2 | Modelling of Electro-Ribbon Actuators considering stiffness variation.** **a** Schematic of the actuator's deformation in passive and active states (10 kV) under 18 mm extension. The simulated beam dimensions are 12.7 mm wide, with normalized effective stiffness $EI_{eff}/w$ = 0.002 Nm (first line) and $EI_{eff}/w$ = 0.008 Nm (second line), respectively. The vacuum permittivity is 8.85 pF m$^{-1}$, the insulator permittivity is 4.62, and the insulator thickness is 130 μm. The liquid has a dielectric constant of 2.7, and air has a dielectric constant 1. **b** Simulation result of the sandwich ERF beam showing reduced displacement under central loading and the corresponding increase in normalized effective stiffness with rising ERF voltage

(see Supplementary Information). The circles represent the stiffness of the three-point bending shown in Fig. (**d**). **c** Simulation result of the force-extension relationship in actuators with varying normalized effective bending stiffness, where solid lines and dashed lines represent force variations at 6 kV and 0 kV, respectively. **d** Three-point bending of an ERF beam. A flexible ERF beam can change bending stiffness under different voltages such as 4 kV and 8 kV. The red part is a support structure, and a 12 g load is placed at the center of the beam after applying voltage. Source data are provided as a Source Data file.

previous studies exploring ERF beam characteristics under small deflection[54], we first investigate the performance of thin ERF sandwich beams under large deformation. According to Supplementary Equation (1), the bending stiffness of a sandwich beam is related to the thicknesses of the face sheet and core[55]. Therefore, we investigate the influences of electrode thickness, ERF layer thickness, and beam

length, respectively. To compare the stiffness variation, the equivalent bending stiffness $(EI)_{eq}$ is calculated as $(EI)_{eq} = mgL^3/3y$.

Figure 3b shows the beam displacement for electrode thicknesses of 50/50, 50/30, and 30/30 μm under a 2 g load. As electrode thickness increases, the bending stiffness $EI$ also increases, where $I$ is proportional to the electrode's thickness and width. At similar stiffening

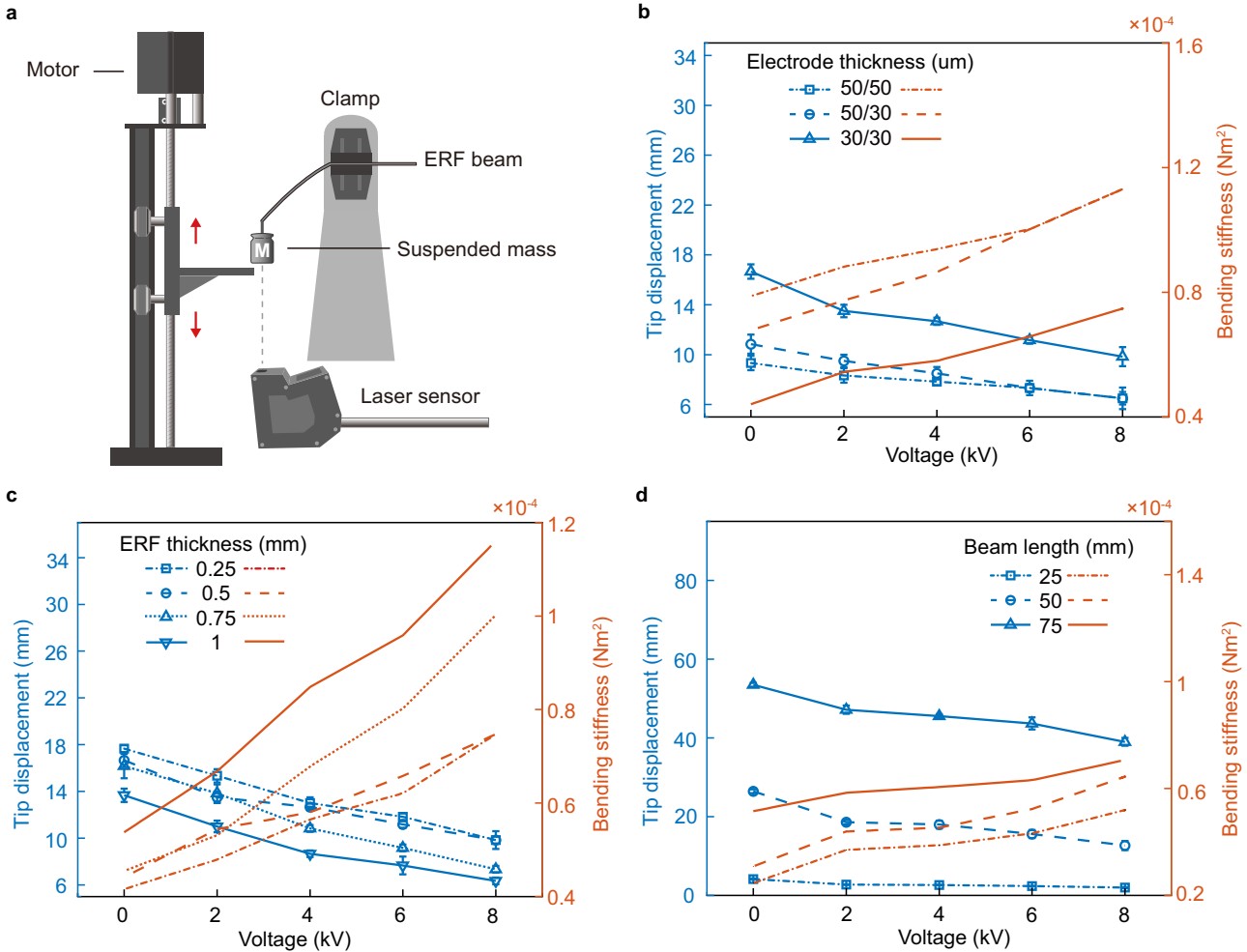

**Fig. 3 | Evaluation of stiffness variation properties for Electrorheological Fluid (ERF) beam. a** Schematic view of the setup used to measure the ERF beam deflection; Comparison of deflection and bending stiffness under different stiffness voltages for different (**b**) electrode thickness, (**c**) ERF layer thickness, and (**d**) beam length. The blue line represents beam tip displacement (left axis), and the red line represents bending stiffness (right axis). The error bars indicate the standard deviation between 3 trials. Source data are provided as a Source Data file.

voltages, thinner electrodes show larger deflections because of the reduced stiffness. With voltage application, the ERF's shear modulus increases due to the electric field. Higher voltages imply higher yield stress within the ERF, reducing the beam deflection for a given load. This voltage-deflection relationship was consistent across experiments. For example, under a 2 g load with an 8 kV increase in stiffening voltage, the equivalent bending stiffness for the 50/50, 50/30, and 30/30 μm electrodes rose by 43.59%, 66.67%, and 69.49%, respectively. The increases for a 6 g load were 40.82%, 25%, and 29.22% (Supplementary Fig. 1a). Notably, the 30/30 μm configuration showed a broad stiffness variation range under varied loads, highlighting the improved actuator stiffness control with thinner electrodes.

Figure 3c presents the displacement for beams with different ERF layer thicknesses under a 2 g load. Displacement consistently decreases with rising ERF layer thickness and stiffening voltage. Under a 2 g load, the stiffness for beams with 0.25, 0.5, 0.75, and 1 mm ERF thickness increased by 79.66%, 69.49%, 120.46%, and 115.79%, respectively. With a 6 g load, the rises were 25.15%, 29.22%, 33.79%, and 29.58% (Supplementary Fig. 1a). The results suggest that ERF beam stiffness, influenced by the applied loads and deformation, becomes more significant at smaller loads. For example, under a 2 g load, beams with 0.25- and 0.5 mm layers nearly doubled in stiffness, while the 0.75- and 1 mm layers experience over a two-fold increase. Greater ERF thickness

implies higher inherent stiffness. Yet, a notable deformation in beams suggests a decreased stiffness variation capacity.

Figure 3d shows the impact of beam lengths on actuator equivalent stiffness, which shows deformation and stiffness increases with the beam length. A longer beam exhibits smaller stiffness variations compared to a shorter one, especially as applied loads rise from 2 g to 6 g (Supplementary Fig. 1). Beam theory attributes this to increased nonlinearity in longer beams under tip loads. Experimentally, a 25 mm beam showed limited deformation but considerable stiffness variation. This implies a trade-off in ERF beam-based actuator design: while longer beams offer a larger deformation range, shorter beams provide more significant stiffness variations but limited displacement due to their length.

## Characterization of ESRA

Figure 4 shows the isometric and isotonic characterizations of the variable stiffness electro-ribbon actuator. Based on the test results of the ERF beam, the specifications for the ESRA actuators have been chosen as follows: 30/30 μm electrodes, 0.5 mm ERF thickness, and a total length of 100 mm. Figure 4c illustrates the actuation stroke and time when lifting a load of 8 g, while the results of lifting loads of 4 and 12 g are shown in Supplementary Fig. 2a. In this setup, the actuator behaves like an active tension spring. As such, the amount of initial

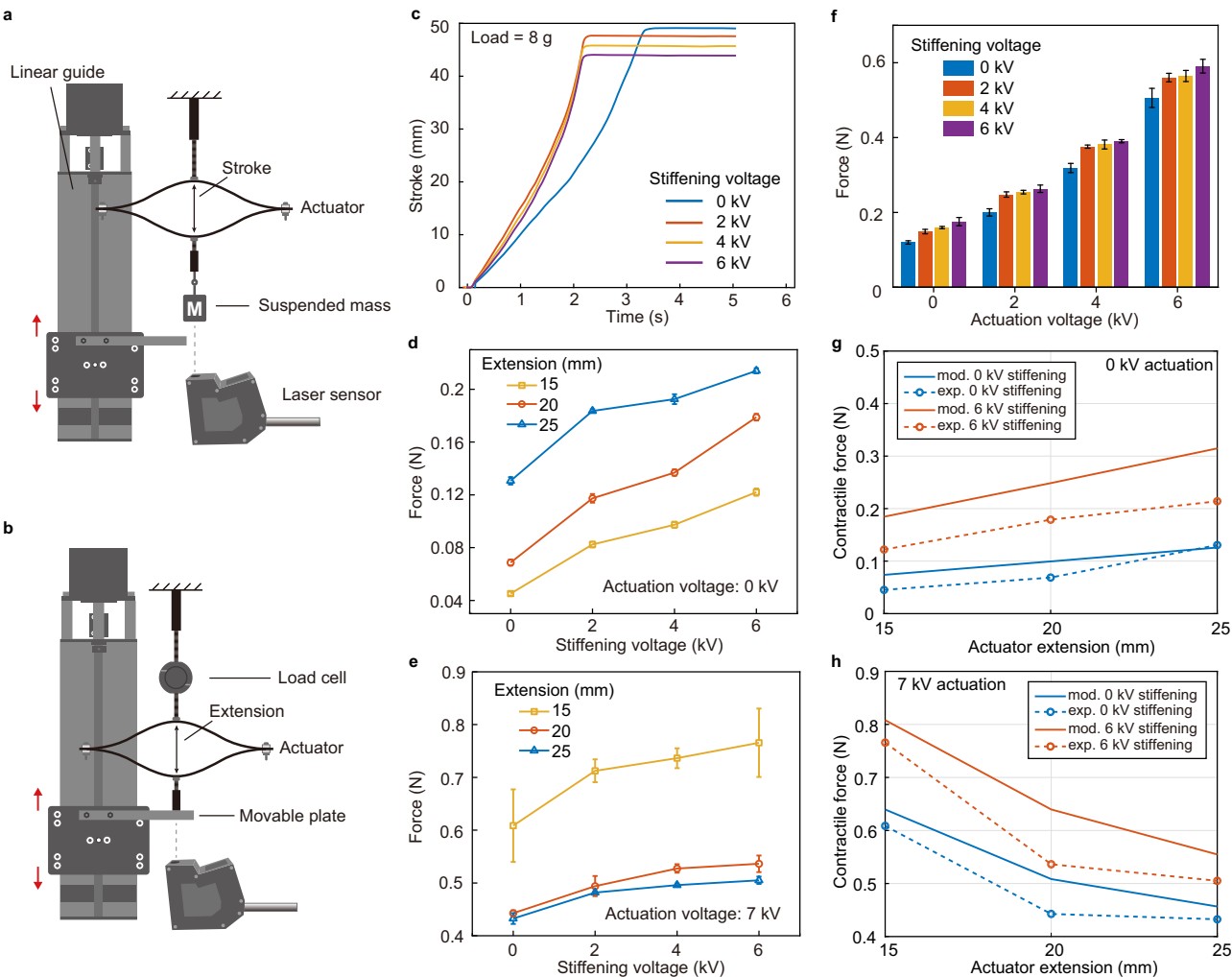

**Fig. 4 | Performance characterization of ESRA. a** Test protocol for isotonic configuration and (**b**) for isometric configuration. **c** Comparison of stroke of an actuator with 30/30 um electrodes, 0.5 mm ERF thickness, and 7 kV actuation voltage, lifting loads of 8 g under four different stiffening voltages. **d** Isometric contractile force at 15-, 20-, and 25 mm actuator extension when actuation voltage is 0 kV. **e** Isometric force under 7 kV actuation. **f** Contractile force at 18 mm under actuation and stiffening voltage from 0 to 6 kV. The error bars indicate the standard deviation between 3 trials. **g**, **h** Comparison of modelling and experimental results under 0 and 6 kV stiffening voltages when actuation voltages are 0 kV (**g**) and 7 kV (**h**). Source data are provided as a Source Data file.

extension exerted by the load on the actuators increases monotonically with the applied loads. A shorter extension corresponds to a smaller angle at the zipping point, resulting in a stronger electric field and, therefore, larger electrostatic force and faster zipping speed. Under identical loads, applying stiffening voltage decreases both the actuation stroke and time. A comparative analysis of the stroke under different stiffening voltages reveals interesting trends: for example, under a 12 g load and applying a 6 kV stiffening voltage, the stroke can be reduced to the equivalent stroke observed under an 8 g load with zero stiffening voltage (Supplementary Fig. 2b). Moreover, the reduction in actuation time becomes more noticeable with increasing loads: the actuation time can be halved under a 12 g load when stiffening voltage is applied (Supplementary Fig. 2a). We observed a non-monotonic decrease in contraction time with increasing voltage after applying the stiffening voltage. For instance, the lifting speed at 6 kV under a 12 g load is marginally less than that observed at 2 kV. This behavior could be attributed to the non-uniform distribution of the ERF suspension within the bean induced by a large deformation.

Figure 4d, e presents the contractile force of the actuator at three fixed extensions in both unactuated and actuated states. When zero actuation voltage is applied, the contractile force of the actuator

increases with extension, consistent with the behavior of a tension spring mechanism. As the stiffening voltage increases, the actuator's stiffness (analogous to the stiffness of a spring) increases, resulting in a corresponding increase in the tensile force. For instance, at a 15 mm extension, the force rises from 0.045 N at 0 kV stiffening voltage to 0.122 N at 6 kV stiffening voltage, which is significantly higher than the force of 0.069 N observed at a 20 mm extension and is close to the force of 0.131 N measured at a 25 mm extension without the applied voltage. To quantify this change further: At 15 mm, 20 mm, and 25 mm, the force increases by 171.11%, 160.26% and 63.86%, respectively, when the stiffening voltage rises from 0 kV to 6 kV. It shows that ERF significantly amplifies the static tensile force of the actuator, which can be dynamically modulated by the stiffening voltage.

When an actuation voltage of 7 kV is applied, a notable enhancement in the contractile force is observed (Fig. 4e), which can be attributed to the electrostatic attraction between the electrodes. Contrary to the behavior observed in the unactuated state, the actuator demonstrates a higher contractile force at shorter extension distances. A shorter extension correlates with a reduced zipping angle, thereby facilitating the zipping behavior and resulting in enhanced electrostatic force. Analogous to the unactuated state, applying a

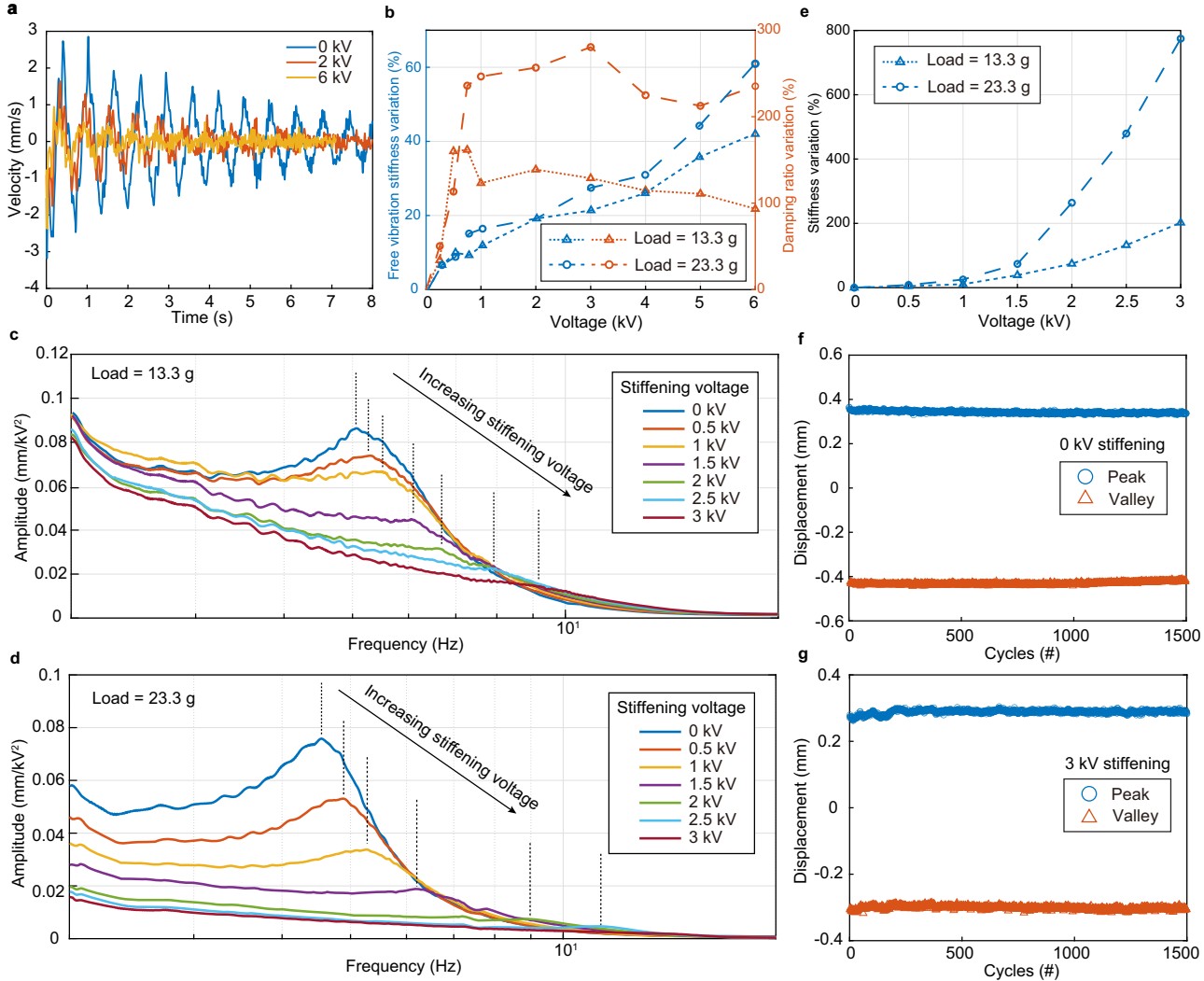

**Fig. 5 | Dynamic characterization of ESRA. a** Velocity of a free-falling mass over time under different stiffness voltages where the mass is 23.3 g. The peak velocity amplitude at 0 kV is significantly larger, and the oscillation lasts longer compared to when a stiffening voltage is applied, which suppresses the oscillation. **b** Variation in effective stiffness and damping ratio under different stiffening voltages during the free response for 13.3 g and 23.3 g loads. **c** Frequency response of an actuator with different stiffening voltages under a chirp signal for actuation, with an applied load of 13.3 g. Vertical dashed lines represent the position of peak amplitude for each response curve. **d** Frequency response with an applied load of 23.3 g. **e** Stiffness variation in the frequency response, calculated from the resonance frequencies shown in panels (**c, d**). **f** Durability test of ESRA, showing the displacement of an actuator lifting 17 g with a bipolar signal of 5 kV at 2 Hz without stiffening. Circles and triangles represent peak and valley values of displacement, respectively. **g** Durability test with a 3 kV stiffening voltage, while blue and red circles represent peak and valley values of displacement, respectively. Source data are provided as a Source Data file.

stiffening voltage was found to further augment the contractile force of the actuator. This finding suggests the capability to dynamically modulate the contractile force by simultaneously adjusting both the actuation and stiffening voltages. As illustrated in Fig. 4f, we analyzed the contractile force at an extension of 18 mm with both actuation and stiffening voltages from 0 kV to 6 kV. Under these conditions, a stepwise increment in contractile force was observed, ranging from 0.12 N to 0.59 N. Compared with conventional electro-ribbon actuators, the ESRA demonstrates a significantly enhanced ability to adjust the contractile force, owing to its inherent variable stiffness characteristics.

Dynamic tests were conducted to further study the ESRA by analyzing changes in stiffness and damping through the oscillations generated during the free response and frequency response. In a free response test, the actuator lifts a heavy object until it reaches maximum contraction, after which the actuation voltage is suddenly removed, allowing the actuator to fall freely. Figure 5a illustrates the

velocity response of the actuator oscillating under a load of 23.3 g at 0, 3, and 6 kV. At 0 kV, the actuator's stiffness and damping are relatively low, resulting in high peak speeds and continuous oscillations, as it cannot effectively dissipate kinetic energy. Upon applying stiffening voltage, both stiffness and damping are enhanced, improving the actuator's ability to dissipate oscillation energy. This results in reduced peak speeds and curtailed oscillations, which is beneficial for achieving precise positioning without overshoot and for maintaining stability against external disturbances. Figure 5b demonstrates the changes in stiffness and damping ratios, calculated from the oscillation period, in response to varying stiffness voltage. With increasing applied voltage, the stiffness correspondingly rises, this increased free vibration stiffness indicates enhanced resistance to large disturbance. The damping ratio initially increases and then stabilizes. With a 13.3 g load, the damping ratio peaks at ~ 500 V before leveling off. When a 23.3 g load is applied, the damping ratio sharply increases up to 1 kV, after which the rate of increase diminishes significantly.

According to our approximate dynamic model (see the dynamic model in Supplementary Information), the actuation force generated upon applying the actuation voltage acts as an additional spring unit, with its spring stiffness influenced by the stiffening voltage. We use a frequency actuation signal to characterize the actuator by analyzing its frequency response. There is no significant amplitude peak when the variable stiffness voltage exceeds 3 kV under the same value of actuation voltage (Supplementary Fig. 4); therefore, we limit our frequency response calculations to the 0–3 kV range. Figures 5c and d show the frequency responses under 13.3 g and 23.3 g loads, respectively. At low frequencies, the amplitude decreases as the voltage increases, indicating enhanced stiffness. The resonance frequency— the point where the amplitude significantly increases—shifts higher as the voltage increases, reflecting a corresponding increase in stiffness. Figure 5e quantifies the equivalent stiffness based on the resonance frequency, revealing that stiffness under the actuation voltage increases with the stiffening voltage, with the rate of increase accelerating. Specifically, at 3 kV, the stiffness at loads of 23.3 g and 13.3 g increases by approximately 800% and 200%, respectively. These dynamic tests reveal that at lower stiffening voltages, the structure's damping can be enhanced while maintaining stiffness within a narrow range, whereas, at higher voltages, stiffness can be adjusted while the damping ratio remains relatively stable.

## Muscle-like functions

We demonstrate an application of the ESRA in the control of system stiffness and damping under shock and disturbance. Figure 6 shows a basic robotic arm supported by a single ESRA prototype in two states: passive (Fig. 6a and b) and actuated (Fig. 6c). Subjecting the arm to an initial displacement with a rope and then releasing the rope, the arm exhibits oscillatory behavior at a natural frequency of 3.7 Hz (see Fig. 6a and Supplementary Movie 1). Applying a 7 kV stiffening voltage modifies the arm's damping and stiffness, raising its natural frequency to 4.6 Hz, reducing peak amplitude to 58%, and shortening oscillation time by more than 50%. Figure 6b shows the dynamic response in impact damping and stiffness of the robotic arm when is hit by an external object (Supplementary Movie 1). When an 8.8 g weight is dropped onto the arm, inducing free fall impact, it oscillates for ~ 2.2 s with a peak amplitude of around 50 mm. In contrast, applying a 7 kV stiffening voltage in this scenario augments the stiffness and impact damping, reducing the oscillation amplitude by 41.44% and shortening the oscillation time to less than 0.8 s.

Figure 6c demonstrates the actuation capabilities of the ESRA and its responses once the arm hit by an external object (Supplementary Movie 2). In scenarios involving substantial impact on the arm, it fails to withstand the impulse adequately when not subjected to stiffening. Consequently, the tip of the arm experiences a significant drop (approximately − 30 mm), followed by a gradual oscillatory rise. In contrast, applying a 7 kV stiffening voltage significantly attenuates the displacement response caused by the falling weight, preventing negative displacements and facilitating peak displacement achievement. Without stiffening, the arm does not reach this peak within 15 s due to excessive fall-induced displacement. Figure 6c. (ii) shows the response to a smaller weight impact, where increased stiffness via a 7 kV voltage results in significantly reduced oscillations compared to the unstiffened state. Overall, the ESRA's enhanced stiffness and damping properties improve its resistance to impact and disturbances in both passive and active states.

ESRA presents promising applications as artificial muscles for prosthetics and wearables, enhancing physical human-robot interaction mobility restoration and augmentation (Supplementary Movie 3). A single-degree-of-freedom (1-DOF) musculoskeletal model is developed to demonstrate this application. Figure 7 shows three operating modes of ESRA: (1) standard actuation, (2) actuation with stiffness enhancement, and (3) resistance enhancement. In the standard

actuation mode (Fig. 7a), ESRA functions as a muscle with over 90% contractions through voltage application, deflecting the arm by approximately 70 degrees. In the stiffness-enhanced actuation mode (Fig. 7b), ESRA augments payload capacity and sustains loads without requiring a voltage increase. For example, when a weight is disengaged at a height of 108 mm, an arm in a non-stiffened state falls under an 11.3 g load at the palm, whereas in a stiffened state, it can sustain and gradually elevate the load to a final position of 120 mm.

In the resistance enhancement mode, ESRA remains unactuated, functioning as a passive spring with adjustable mechanical impedance (Supplementary Movie 4). Figure 7c illustrates an antagonistic ERA-ESRA muscle pair, with ERA serving as the ventral actuator and ESRA as the dorsal resistance element in the upper arm. Comparing conditions of non-stiffened ESRA and ESRA stiffened at 7 kV, it is evident that under a 6 kV ERA actuation voltage, the stiffened arm ascends more slowly and fails to reach the peak displacement within 18 s, unlike the non-stiffened state, which peaks at 14 s and sustains this position. These tests demonstrate the potential use of the ESRA for complex multi-mode transformations.

## Dual actuation systems with adjustable stiffness and damping

Multiple actuators in configurations like muscle-muscle (Fig. 8a) or muscle-tendon (Fig. 8d) connections can be employed to regulate varying stiffness and damping levels. Firstly, we present an actuation system that serves as a one-dimensional analogue to the human arm's agonist-antagonist muscle system. As shown in Fig. 8a and b, this system comprises two components: the ESRA and the ERA. The ESRA can be modeled as a spring-damper system with a nonlinear spring coefficient, denoted as $k_{mat}$ due to the material properties and $k_{ES}$ due to the electrostatic force, along with a nonlinear damping ratio $c$. The ERA, which exhibits relatively high stiffness and negligible damping compared to the ESRA, is modeled as a contraction unit and a spring unit. These two components are linked by a point mass, with both the upper and lower springs initially extended (detailed in the equivalent model provided in the supplementary information).

The ERA acts as an exciter, providing stimuli to the system and maintaining high stiffness to inhibit movement initiated by the ESRA. Figure 8b outlines three basic modulation modes of ESRA: (1) Low stiffening voltage: Applying a low stiffening voltage to the ESRA enhances damping while minimally affecting stiffness. (2) High stiffening voltage: Applying a substantial stiffening voltage to the ESRA significantly increases both stiffness and damping. (3) Low actuation voltage: Applying a minimal actuation voltage to the ESRA, insufficient for actuation, results in negligible displacement but increases structural stiffness. A hybrid mode can be achieved by simultaneously applying both stiffening and actuation voltages. Figure 8c shows the frequency response when a sinusoidal excitation signal is applied to the ERA, with the signal frequency progressively increasing over time. The left plot displays changes in relative damping and stiffness. In the mode of ($V_{Act.}$ OFF, $V_{Stiff.}$ ON(low)), there is a notable decrease in amplitude along with a slight increase in resonance frequency, indicating higher damping with minimal change in stiffness. In contrast, the mode of ($V_{Act.}$ ON, $V_{Stiff.}$ OFF) shows a reduction in amplitude along with a shift to a higher resonance frequency, suggesting increased stiffness and low damping—an effect similar to muscle co-contraction[56]. In the mode of ($V_{Act.}$ OFF, $V_{Stiff.}$ ON(high)), vibrations are significantly suppressed, indicating both high damping and stiffness.

When multiple ESRAs are connected in series, they mimic a muscle-tendon system by enabling the regulation of equivalent stiffness and damping. We formed a system by connecting two ESRAs in series, analogous to a pair of Voigt elements (Supplementary Fig. 16), and subjected it to a load as depicted in Fig. 8c. As shown in Fig. 8e, when a stiffening voltage $V_{S1}$ is applied to the upper actuator, the

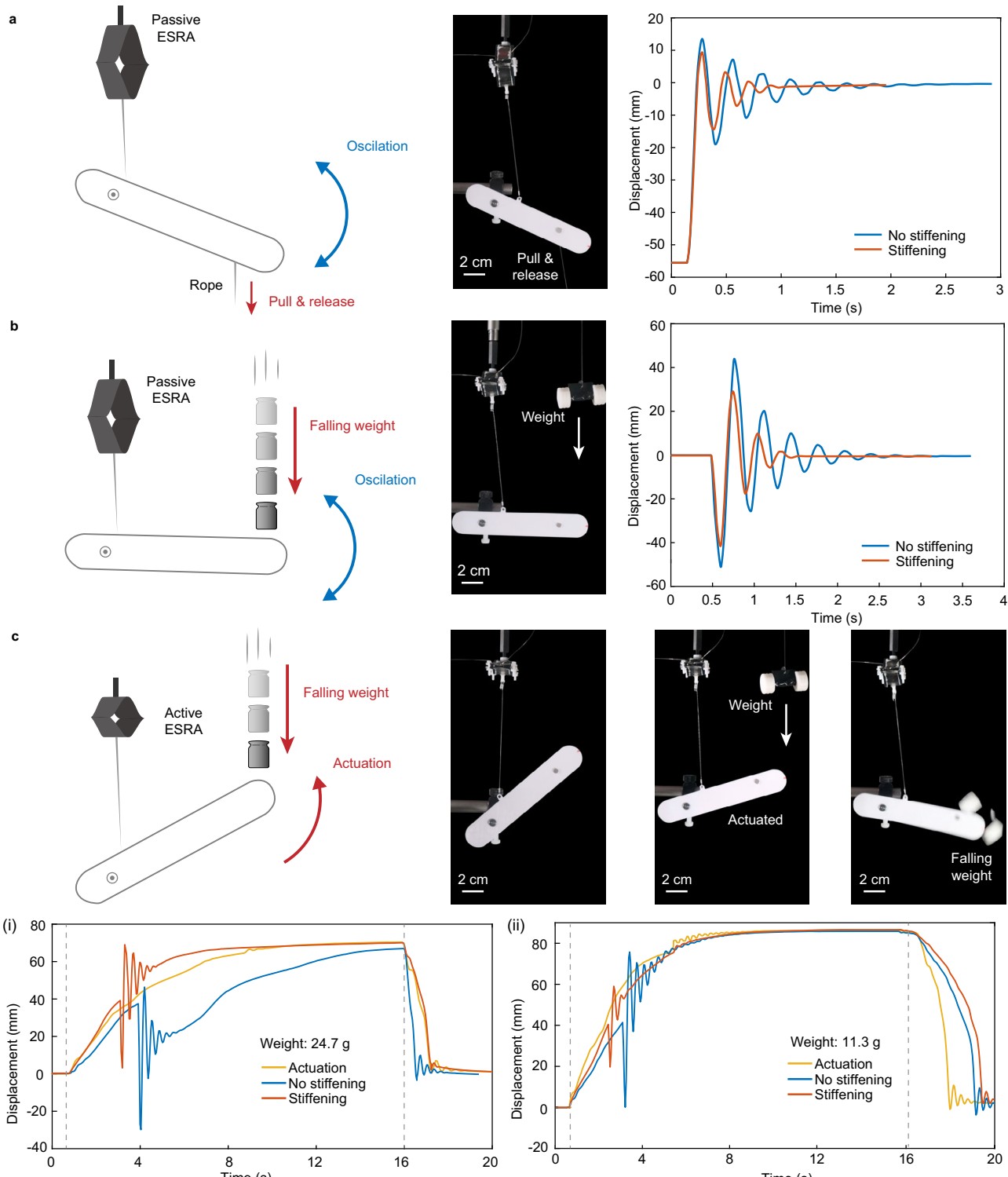

**Fig. 6 | Actuation and impact resistance of an Electro-Stiffened Ribbon Actuator (ESRA)-supported arm. a** Passive operation of the system with variable stiffness. The arm was supported by an ESRA undergoing initial displacement, pulled down by a rope, and subsequently released. The displacement of the arm's tip is shown with blue and red lines, representing 0 kV and 7 kV stiffening voltages, respectively. **b** Passive operation of the system with variable stiffness impacted by a free-falling 8.8 g object onto the arm's tip. The blue and red lines show the arm tip's displacement with 0 kV and 7 kV stiffening voltages, respectively. **c** Active operation of the system, hit by released (i) 24.7 g and (ii) 11.3 g objects, respectively. The yellow line represents the actuation displacement under normal conditions without external impact, while the blue and red lines illustrate the displacement change under external impact in unstiffened and stiffened states, respectively. The vertical dashed lines indicate the moments when the voltage is turned on and off. Source data are provided as a Source Data file.

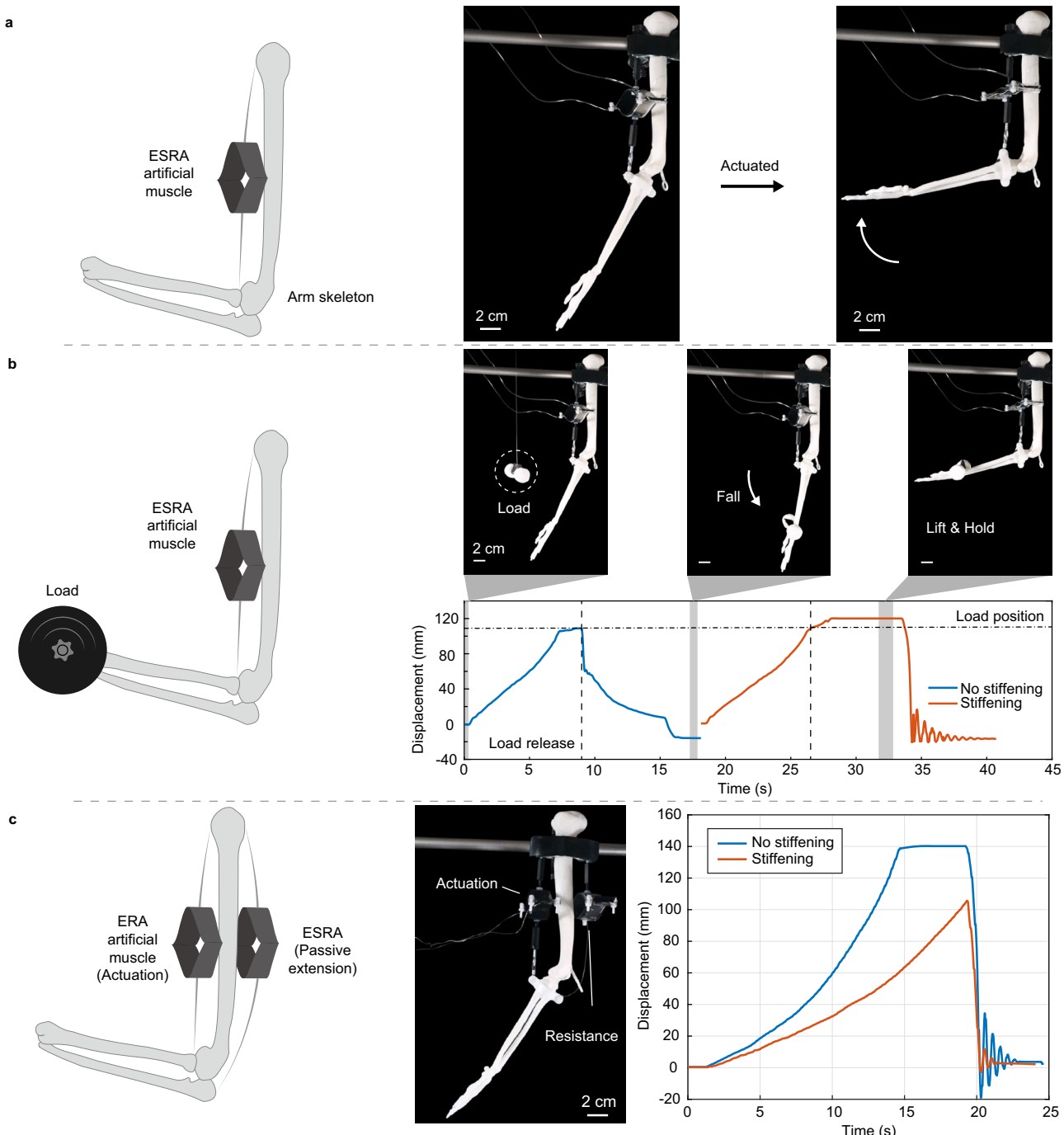

**Fig. 7 | 1-DOF musculoskeletal model based on Electro-Stiffened Ribbon Actuator (ESRA). a** Standard actuation mode of ESRA, functioning as an Electro-Ribbon Actuator (ERA) muscle to flex the arm. **b** Stiffness-enhanced actuation mode: an 11.3 g load is positioned at 108 mm and released as the arm attains this height. Palm motion under non-stiffened and stiffening states is represented by blue and red lines, respectively. The horizontal dashed line indicates the load position, the vertical dashed lines mark the load release time, and the grey area represents the three states shown above the plot. **c** Resistance enhancement mode: the ventral ERA in the upper arm serves as an actuator for arm elevation, and the dorsal ESRA as a resistance element. Palm motion under non-stiffened and stiffening states is indicated by blue and red lines, respectively. Source data are provided as a Source Data file.

system's overall stiffness increases while damping decreases. Conversely, applying a stiffening voltage $V_{S2}$ to the lower actuator enhances both stiffness and damping. We assessed this variable stiffness and damping capability through frequency response analysis. As Fig. 8f shows, with $V_{S2}$ held at 0 kV, increasing $V_{S1}$ from 0 to 5 kV raises the system's resonant frequency and equivalent stiffness from 7.5 to 12.5 Nm and lowers the damping ratio relative to excitation force from 0.55 to 0.36. For a constant $V_{S1}$, when the stiffening voltage $V_{S2}$ changes

from 0 kV to 5 kV, both stiffness and damping increase simultaneously. By independently modulating the two voltages, we can achieve diverse combinations of stiffness and damping, with the total variations reaching up to 2.2-fold for stiffness and 2.6-fold for damping, respectively. As demonstrated in Fig. 8a (right) and further shown in Supplementary Movie 5, at an excitation frequency of 5 Hz, the vibration amplitude can be increased by 90% with high stiffness settings, or suppressed by 45% with elevated damping.

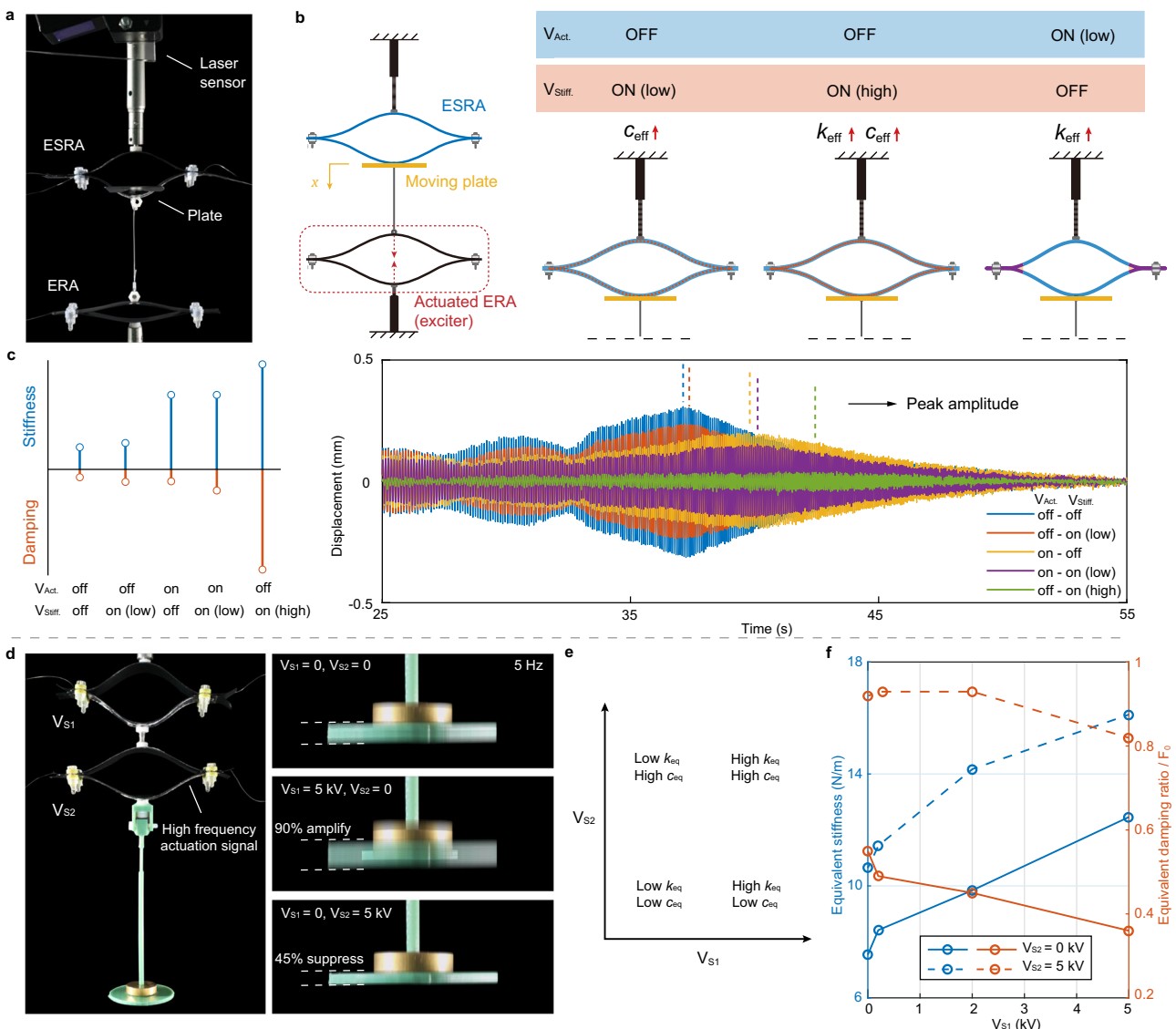

**Fig. 8 | Dual actuators. a** Experimental setup featuring a laser displacement sensor to monitor the movement of the plate, which links the Electro-Stiffened Ribbon Actuator (ESRA) and the Electro-Ribbon Actuator (ERA), with high actuation voltage applied to the ERA, acting as an exciter. **b** Schematic representation of the ESRA-ERA agonist-antagonist system in one dimension, including three fundamental operating modes. **c** Variation in stiffness and damping, along with system responses under different ESRA voltage conditions, when a chirp signal actuates the ERA. The right plot begins at 25 s, and vertical dashed lines indicate the position of the peak amplitude for each condition. When actuation voltage is ON, $V_{Act.} = 2000$ V; when the stiffening voltage is ON (low), $V_{Stiff.} = 250$ V; and when the stiffening voltage is ON (high), $V_{Stiff.} = 4000$ V. The left figure shows relative changes in stiffness and damping. **d** Dual ESRA configuration in series, lifting a load of 15.6 g. A high-frequency actuation signal stimulates the lower actuator, acting as the exciting force, with $V_{S1}$ and $V_{S2}$ indicating the stiffening voltages for the upper and lower actuators, respectively. Vibration under a 5 Hz excitation is shown under two different conditions when stiffness is dominant: $V_{S1} = 5$ kV and $V_{S2} = 0$ kV; and damping is dominant: $V_{S1} = 0$ kV and $V_{S2} = 5$ kV. **e** Variation in stiffness and damping in response to changes in $V_{S1}$ and $V_{S2}$. **f** Changes in equivalent stiffness and damping as functions of $V_{S1}$, where solid line represents $V_{S2} = 0$ kV and the dashed line represents $V_{S2} = 5$ kV. Source data are provided as a Source Data file.

## Discussion

The Electro-Stiffened Ribbon Actuator (ESRA) introduced in this paper demonstrates the seamless integration of a lightweight variable stiffness technology into the electrostatic actuator, for building a robotic muscle with a similar driving and control unit. By incorporating the Electrorheological Fluid, the ESRA offers variable viscoelastic properties in both passive and active modes, achieving approximately an 8-fold increase in stiffness and a 3-fold increase in damping. The ability to modulate stiffening and actuation voltages enhances the force output range − a 2.5-fold increase in passive force and a 1.5-fold increase in the active force range when stiffening voltage is applied. The increased stiffness and damping improved dynamic alteration capabilities, showcased by quicker lift periods in the stiffened state

and improved resistance to impacts and disturbances. Moreover, its inherently simple design offers exceptional adaptability, enabling replication of simultaneous stiffness control in actuators by replacing electrodes with a thin ERF-enclosed beam across various electrostatic flexural actuators from finger-like bending[57] and ratcheting antiphase actuators[58] to flappy wing[33], wearable pumps[34] and origami structures[59].

ESRA demonstrates performance in variable stiffness and damping that is comparable with the advanced technologies (Supplementary Table 1), while retains key advantages to ERA[35], such as a lightweight design and ultra-high deformation. In comparison to alternative variable stiffness mechanisms, such as layer jamming[23] and low melting point alloys[24], ESRA offers superior power efficiency and

rapid response times, comparable to other electric solutions like electrostatic clutches[37,38,60]. Under similar configuration, the degree of stiffness variation we achieve under large deformation notably exceeds that of electrostatic clutches (Supplementary Fig. 17). ESRA features a compact design with monolithic integration while offering an alternative flexible solution to enhance the actuator's load capacity and dynamic response.

The performance and adaptability of the ESRA demonstrated in this study highlight its suitability for a diverse range of applications. Its inherent flexibility, silent operation, lightweight design, and variable stiffness allow for its use in future wearable assistive technologies. In robotics, the capacity for stiffness variation enhances compliance for soft interfaces and increases resistance to external forces in conditions requiring high stiffness. Improved damping also allows the ESRA to quickly mitigate impact and disturbances, making it advantageous in robots' operation in varying environments. The capability to stiffen without limiting the actuation range of a soft actuator improves the load-bearing capacity of robotic arms and wearable devices, thereby facilitating greater load adjustment and operational efficiency. The adjustable resistance feature can also be used in rehabilitation wearables, such as robotic sleeves, facilitating effective strength training by allowing dynamic resistance modification, while retaining the capability to switch functions and assist patients in arm movement as required. In addition, linking multiple ESRAs can create antagonistic muscle structures or muscle-tendon structures, allowing for various stiffness and damping combinations by independently adjusting the stiffening voltages of different actuators. As an application in vibration control, compared to conventional variable stiffness and damping vibration control devices, the ESRA system offers a similar range of adjustments with great simplicity (Supplementary Table 2), while still retaining robust actuation capability. This capability to adjust stiffness, along with tunable resonant frequencies and damping enables new uses, from enhancing the comfort and performance of suspension systems to augmenting the speed and efficiency of leg-based robotic locomotion.

ESRA's stiffness and damping capabilities can be enhanced with stronger ERF while adding more stacking layers allows for greater independent control. Our model shows that employing liquid dielectrics with higher dielectric constants and matched insulation layers with enhanced breakdown strength improves actuation performance. In conclusion, ESRA mirrors the functionality of biological muscles in different aspects, underscoring its potential for developing compliant bionic robots.

## Methods

### Fabrication of ESRA
The ESRA consists of two identically configured flexible beams, resulting in a symmetric design, as illustrated in Supplementary Fig. 5a. Each beam comprises dual flexible strip electrodes, separated by a gap filled with ERF. This fluid reacts to the electric field applied between the electrodes. One electrode is sandwiched between two insulation layers: one layer insulates it from the ERF, while the other insulates the electrode for electrostatic zipping. The second electrode in the beam is externally sealed using flexible sealing tape. To facilitate electrostatic zipping, rigid clamps are employed at both ends of each beam, forming two distinct zipping angles. The actuator can be extended by applying a central load. A liquid dielectric bead is added at each hinge, and the applied high voltage produces electrostatic force to cause zipping and contraction. The ESRA is flexible and simplistic in design (Supplementary Fig. 5b).

### ERF beam characterization
Figure 3a shows the experimental setup used to characterize the variable stiffness property of the ERF beam. We clamped the ERF beam to form a cantilever beam structure with the end of the beam connected to a known mass. A stepper motor (NEMA17 ACME Lead Screw Actuator, Ooznest, UK) was used to support the beam to level at the initial position, and then the beam was released by the motor to bend freely. A laser displacement sensor (LK-G40152, Keyence, Japan) was used to measure the displacement of the beam's tip.

### ESRA characterization
Figure 4a and b show the experimental setup of isotonic and isometric tests to characterize the performance of ESRA. For the isotonic test, the upper central point of the actuator was securely connected to a rigid frame as shown in Fig. 4a. A known mass was attached to the lower central point of the actuator, imposing a constant vertical load and thereby causing the actuator to extend under this constant force. A laser displacement sensor was used to monitor and record the displacement of the load as well as the time taken for the actuator to contract.

During the isometric test, an initial voltage was applied at a 2 mm extension, and the actuator was stretched to various extents using a stepper motor connected at its lower central point, as depicted in Fig. 4b. The upper central point of the actuator was connected to a load cell (DBCR-10N-002-000, Applied Measurements Ltd., UK), which was attached to a stable fixing bracket. The high voltage applied generated an electrostatic attraction, resulting in a pulling force at the center of the actuator. This force was quantified using the load cell. In each test, a drop of 50 cSt silicone oil (Sigma-Aldrich, USA) is added to the two hinges of the actuator using a pipette.

The dynamic test was conducted using the experimental setup shown in Fig. 4a. In the free response test, a 4 mm thick nut was attached to the ESRA's upper beam to prevent the actuator from closing completely which would make its detachment slower. A large voltage was applied to achieve maximum actuator contraction, bringing the lower beam into contact with the nut. Then, the voltage was suddenly removed, and the actuator released under the weight's gravity. The displacement of the mass was measured by the laser sensor. During frequency testing, different stiffening voltages were applied to the actuator. The chirp signal was then applied, and the displacement of the mass was measured.

### Durability test
The durability test was conducted using the experimental setup illustrated in Fig. 4a. Figure 5f and g and Supplementary Fig. 6 show the results from over 1500 cyclic tests performed at 2 Hz with attached weights of 17 g. These tests were driven by bipolar signals supplied by high-voltage laboratory power supplies (UltraVolt HVA series), where the applied voltage was set to 5 kV. During the durability test, there was no significant change in strain with 0 kV stiffening voltage. At 3 kV stiffening voltage, the amplitude baseline shifts slightly upward before stabilizing, with no significant change in amplitude.

### One-dimensional agonist–antagonist muscle system
The ESRA and ERA were connected vertically by a nylon rope, with the upper end of the ESRA fixed and the lower end of the ERA fixed, creating an overall length of 17.5 cm. The ERA was fabricated using electrodes 12.7 mm wide, 75 mm long, and 70 μm thick. A moving platform weighing 3 g, was connected in the middle of this assembly. To measure the displacement of the platform, a laser displacement sensor was used.

### ESRA-supported beam
The oscillation displacement of the ESRA-supported arm was demonstrated on a simplified robotic arm beam, measuring 14.5 cm in length, 2.5 cm in width, and 2 mm in thickness (Fig. 6). This beam was mounted 3 cm from its left end to rigid support using bearings, allowing for free rotation. The ESRA unit was installed 10 cm above this beam and connected at a point 5.5 cm to the left of the beam's connection point.

In the demonstration involving ESRA's initial displacement (Fig. 6c), the lower end of the beam was drawn down using a nylon wire to create a 56 mm initial displacement, and then released. Another demonstration focused on the beam's response to impact force (Fig. 6d), where an 8.8 g weight was affixed 8 cm above the beam's right end and then released. In the actuation demonstration (Fig. 6e), the ESRA, in its actuation state, lifted a 12.7 g beam. This was followed by placing and releasing a weight, either 11.3 g or 24.7 g, at the same position used in the impact demonstration.

## 1-DOF musculoskeletal model

A robotic arm bending device was demonstrated on an upper limb musculoskeletal model, featuring an upper arm measuring 13.6 cm and a combined forearm and hand length of 17.5 cm (as depicted in Fig. 7). During the ESRA actuation demonstrations (Fig. 7a and b), the ESRA unit was mounted on a rigid bracket near the shoulder, extending to a point 1.5 cm from the elbow on the forearm. This setup enabled the ESRA-actuation device to lift the forearm, weighing 10.2 grams. In the load lifting demonstration, an 11.3 g load was affixed at the position of 108 mm and was released after the palm reached this height. In a separate demonstration highlighting resistance, the ERA was affixed to the forearm, while the ESRA was mounted on the back of the arm, serving as a passive spring. ERA was fabricated by electrode with 12.7 mm width, 100 mm length, and 40 μm thickness. This configuration allowed the ERA to lift the lower arm by counteracting the contractile force of the ESRA.

## Data acquisition and analysis

In the ESRA dynamic characterization, data acquisition frequency was 10000 Hz. In the demonstration of one-dimensional agonist–antagonist system, data acquisition frequency was 1000 Hz. In the demonstrations of the ESRA-supported beam and 1-DOF musculoskeletal model, the entire process was recorded using a camera (Cannon EOS M50). A video analysis was performed using Tracker software to track and process the displacement data.

## Frequency analysis

Traditional frequency domain analysis techniques are employed for examining frequency responses. A zero-mean signal, such as a pure sine wave, cannot be used because the actuator's response to voltage is strictly positive. Therefore, the system input is defined as:

$$V(t) = V_{DC} + V_{AC}(t) \tag{3}$$

where $V_{DC}$ is the DC offset voltage and $V_{AC}$ is a dynamic zero-mean test signal. The test signal is a chirp signal:

$$V_{AC}(t) = A \sin(2\pi f(t)t) \tag{4}$$

where $A$ is the voltage amplitude, $t$ is time, and $f(t)$ is the frequency, which varies with time as expressed by:

$$f(t) = \frac{f_{end} - f_0}{t_{Test}} \cdot t \tag{5}$$

Here, $f_0$ is the start frequency, $f_{end}$ is the end frequency and $t_{Test}$ is the test duration.

In the ESRA dynamic characterization, the voltage amplitude was set at $A = 3.5$ kV and $V_{DC} = 3.5$ kV. In the demonstration of one-dimensional agonist–antagonist system, the voltage amplitude for ERA was $A = 2.5$ kV and $V_{DC} = 2.5$ kV (Supplementary Fig. 7). In both cases, the frequency and test duration were $f_0 = 0.5$ Hz, $f_{end} = 20$ Hz and $t_{Test} = 60$ s.

MATLAB was used to generate the signal and feed it into the voltage amplifier. To obtain the frequency response data, a fast Fourier transform (FFT) was performed using MATLAB on both the input (amplifier voltage) and output (laser position measurement). Due to the quadratic relationship between the electrostatic force and the input voltage[50], the input signal is calculated as:

$$U_{in} = (V_{DC} + A \sin(2\pi f(t)t))^2 \tag{6}$$

the unit is in kilovolts (kV). The frequency response of the system is determined by dividing the output FFT by the input FFT.

## Calculation of resonant frequency, effective stiffness and damping ratio

In the free response test of ESRA, the system was modelled as suspended spring-mass-damper system (see the model in Supplementary Information). The system exhibited underdamped behaviour, characterized by oscillations with a gradually decreasing amplitude. For the damping ratio, we employ the logarithmic decrement method. The critical parameter here is the logarithmic decrement $\delta$, calculated as the natural logarithm of the ratio of successive oscillation amplitudes, expressed as

$$\delta = \ln\left(\frac{x_n}{x_{n+1}}\right) \tag{7}$$

where $x_n$ and $x_{n+1}$ are the amplitudes of two successive peaks. To enhance the accuracy of the damping ratio estimation, we compute the average logarithmic decrement:

$$\bar{\delta} = \frac{1}{N-1} \sum_{i=1}^{N-1} \ln\left(\frac{x_i}{x_{i+1}}\right) \tag{8}$$

where $N$ is the total number of peaks extracted from the response. For our calculation, we set $N$ to 5. The damping ratio $\xi$, is then estimated from the average logarithmic decrement using the relation

$$\xi = \frac{\bar{\delta}}{\sqrt{4\pi^2 + \bar{\delta}^2}} \tag{9}$$

For our calculation, we set $n$ to 3. The natural frequency, $\omega_d$, is determined by calculating the average time ($T$) between successive peaks. The damped natural frequency is given by

$$\omega_d = \frac{2\pi}{T} \tag{10}$$

And the undamped natural frequency is obtained using

$$\omega_n = \frac{\omega_d}{\sqrt{1 - \xi^2}} \tag{11}$$

Finally, the stiffness of the system can be calculated using

$$k = m\omega_n^2 \tag{12}$$

In the frequency response test, the resonance frequency $f_0$ was extracted from the position of the peak amplitude of the frequency response curve, and the effective stiffness was calculated by

$$k = m(2\pi f_0)^2 \tag{13}$$

## Statistical analysis

Unless otherwise stated, data points represent mean values of n samples, which are specified in the experiment description, and error bars represent SD.

## Data availability

All data needed to evaluate the conclusions in the paper are present in the paper and the Supplementary Information. The experimental data generated in this study have been deposited in the Figshare repository (https://doi.org/10.6084/m9.figshare.28188206). Source data are provided in this paper.

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

## Acknowledgements

We thank NATURABIOMAT GmbH for providing NATURAPACKAGING (BP) films. The work of J.W. was supported by the China Scholarship Council under Grant 202206280075. The open access fee was paid from the Imperial College London Open Access Fund.

## Author contributions

Y.X. contributed conceptualization, methodology, investigation, visualization, and original draft writing. J.W. contributed investigation. E.B. and M.T. contributed conceptualization, methodology, supervision and draft review and editing.

## Competing interests

The authors declare that they have no competing interests.
