## [Transparent Peer Review file · Nature Communications]

Monolithic Electrostatic Actuators with Independent Stiffness Modulation

Corresponding Author: Dr Majid Taghavi

Version 0:

Reviewer comments:

Reviewer #1

(Remarks to the Author)

In this manuscript, Xu et al. developed an electrostatic actuator with independent stiffness controls. The design concept was introduced, the fabrication processes were described and the actuation and stiffness modulation performance were characterized extensively. Overall, this paper demonstrates very clear novelty and is very well written. The reviewer believes that this paper can be potentially publishable in this journal if the following comments can be addressed properly.

Major comments:

1. In this paper, the key concept of independent stiffness modulation (as indicated by the title of this paper) was kept mixing with the impedance (stiffness and damping) modulation (as mentioned in abstract, introduction and in some experimental results). This can be really confusing when going through this paper. By looking at the title, the reviewer felt that 'ok, this paper is about an actuator that can control its stiffness'. Then in abstract (line 26-27), the reviewer felt 'ok, it can also vary the damping ratios'. Later in the results and demonstration parts, the reviewer finally realized that 'the ERF can vary the stiffness and damping simultaneously in this design, but the increase in damping is only a byproduct of the stiffness increase'. So, this sort of flow (as experienced by the reviewer) can potentially lower the impression of this paper, thereby reducing its impacts. As a result, the authors should be very careful with what the 'money shot' of this paper is.
2. To better demonstrate the advantages and novelty of this ESRA design, please compare the key performance indices as well as power consumptions of the ERF adopted in this work as the variable stiffness mechanism and electrostatic force-based clutches and jamming mechanisms.
3. The reviewer noticed that the experimental results are not compared with modelling results in this paper. Please include some modelling results next to the experimental data in the Characterization of ESRA section.
4. Based on Figure 6, both stiffness and damping increase with the stiffening of the ESRA. However, in many circumstances, it might be more desirable to independently control the stiffness and damping of an actuator. Would this be realized by using the ESRA?

Minor comments:

1. Fabrication section can be adjusted to the bottom of this manuscript or in the supplementary.
2. What is the material of the electrodes in Fig. 3?
3. Figure 6 (i & ii), it would be more convenient to have the mass written in the plots.
4. Supplementary Figure 4 A, the legends in the first plot are in wrong colors.

Reviewer #2

(Remarks to the Author)

What are the noteworthy results?

--This paper introduces the "Electro-Stiffened Ribbon Actuator (ESRA)," a new actuator that enhances current "electro-ribbon actuators (ERA)" by adding a variable stiffness function. The authors redesigned the ERA's beam to include a sandwich structure with two electrodes and electrorheological fluid (ERF) in the middle. When voltage is applied between the two electrodes, the viscosity of the ERF increases, thereby stiffening the actuator. This design largely retains the original structure, form factor, and functions of conventional ERAs. Additionally, the stiffening function can operate independently of the actuation, providing greater flexibility. The authors also discovered that the stiffening function could enhance actuation strength, resulting in a higher payload capacity than conventional ERAs.

Will the work be of significance to the field and related fields? How does it compare to the established literature? If the work is not original, please provide relevant references.

--Adding variable stiffness to soft actuators can be significant. Soft robots with variable stiffness have better force transmission, therefore improving power output. The variable stiffness sandwich-structured beam with two electrodes and an ERF infill was developed by Huilan Jing et al., as cited by the author in the introduction. The author has made several modifications to adapt this design for the ESRA. For example, the author uses PVC tape to encapsulate the electrode and VHB tape to seal the ERF, which reduces the beams' thickness, and simplifies the manufacturing process. The authors have also chosen the steel electrode that better suits the ESRA.

Does the work support the conclusions and claims, or is additional evidence needed?

--The author claims that the newly developed ESRA is "compact", "high-contraction", and is a "multifunctional muscle-like actuation solution". This work provided clear figures including actuator dimensions and manufacture process to show that adding variable stiffness does not significantly increase the form factor, and the figures and the supplementary videos show that the compactness of the actuator. Experiment in figure 5 has shown the high-contraction claim. Demonstrations in figures 6 and 7 uses ESRA to simulate muscles and has shown how variable stiffness contributes to the system by comparing the stiffened actuator with the non-stiffened actuator. Overall, the work did support its claims through experiment results.

--The first paragraph of the Results section makes the claim that "ESRA allows for independent modulation of stiffness and damping properties," but the experiments do not seem to justify this claim. In the experiments, stiffness and damping seem to be coupled together, not independently controlled. Further experiments or explanation should be provided to justify this claim or it should be removed.

Are there any flaws in the data analysis, interpretation and conclusions? Do these prohibit publication or require revision?

--I have not found errors in the data analysis and interpretation. All the data explicitly supports the arguments in this work.

Is the methodology sound? Does the work meet the expected standards in your field?

--The characterization of the actuator is well-rounded. I'm interested in the durability of the proposed actuator. It might be helpful to other readers if the authors could hang a weight on the actuator and let it actuate for an extended period, and report the number of cycles the actuator lasts, and the actuation consistency throughout the process.

Is there enough detail provided in the methods for the work to be reproduced?

--Yes, the author has provided enough detail that I am confident that I can reproduce the actuator manufacturing and experiments.

Additional Comments

--The authors can improve the visual presentation of the data. Currently, the color scheme between each figure is not unified. A consistent color scheme across all figures would enhance readability.

--In the plots at the bottom of Fig. 6 it would be easier for the reader to visually compare the responses in the "actuation", "no stiffening", and "stiffening" conditions if they overlapped, rather than plotted sequentially in time.

--In Fig. 7b it is unclear what all of the dashed lines mean. Please clarify in the diagram or caption.

Reviewer #3

(Remarks to the Author)

This work proposed a stiffness-tunable actuator based on electrostatic ribbon actuator and variable stiffness technique of electrorheological fluid.

Honestly speaking, the reviewer cannot see the innovation, significance and contribution of the combination of the two techniques (electrostatic actuator and variable stiffness). Particularly, several previous works have already published the idea of combining soft robots and electrorheological fluid, with the similar fabrication method, e.g., Jing, Huilan, et al. "Variable stiffness and fast-response soft structures based on electrorheological fluids." *Journal of Materials Chemistry C* 11.35 (2023): 11842-11850.

Reviewer #4

(Remarks to the Author)

Version 1:

Reviewer comments:

Reviewer #1

(Remarks to the Author)

The reviewer appreciates the authors' efforts on revising this manuscript by adding dynamic characterizations of the device and adding comparisons of this design against the state-of-the-art. However, the reviewer believes that this paper, in its current form, lacks the substantial significance and impacts to be published in this journal compared with the two electrostatic actuator related papers very recently published in Nature Communications, where the first one being more closely related:

Wang, X., Wang, Y., Zhu, M. et al. 2-dimensional impact-damping electrostatic actuators with elastomer-enhanced auxetic structure. *Nat Commun* 15, 7333 (2024).

Buchner, T.J.K., Fukushima, T., Kazemipour, A. et al. Electrohydraulic musculoskeletal robotic leg for agile, adaptive, yet energy-efficient locomotion. *Nat Commun* 15, 7634 (2024).

As stated by the authors, the key novelty of this paper is focused on the stiffness and damping modulation capability of the electrostatic actuator. However, as is shown in Fig. 5(b), the damping of this device is increased to its peak value at a very low voltage, yet the stiffness of the device increases almost linearly against the voltage, which makes that the stiffness modulation of this devices is almost certainly accompanied by a high additional damping. As the reviewer pointed out in the last round of review, it is more desirable to have individual control of the two properties, especially to be able to publish in such a top tier journal. The authors are suggested to focus more on this aspect if they insist to work on the concept of ES actuator with stiffness and damping modulation capabilities.

Related to the previous comment on the comparison of this design against the state-of-the-art, the authors are suggested to add more comparisons on the stiffness/damping variation ratios. It can be noted from this paper that the stiffness modulation ratio is rather low for this design, so the authors should make strong justifications on why this ER fluid is adopted, instead of using, say, electrostatic layer jamming. Electrostatic layer jamming or clutch has rather rapid response (5 ms) and two layers will be able to increase the stiffness by 4-fold, so it is by no means bulky. See the two papers below.

Hinchet, R. and Shea, H., 2020. High force density textile electrostatic clutch. *Advanced Materials Technologies*, 5(4), p.1900895.

Chen, C., Fan, D., Ren, H. and Wang, H., 2023. Comprehensive Model of Laminar Jamming Variable Stiffness Driven by Electrostatic Adhesion. *IEEE/ASME Transactions on Mechatronics*.

The reviewer suggests the authors to make further major revisions to address the aforementioned comments.

Reviewer #2

(Remarks to the Author)

All of our previous comments have been sufficiently addressed by the authors.

Reviewer #3

(Remarks to the Author)

This work proposes the integration of Electro-Rheological Fluids (ERF) and ribbon electrostatic actuators for multi-mode actuation, including active actuation and stiffening.

Comments:

1. The reviewer expected a novel feature that could not be achieved by either one of the actuation mechanisms alone or through a simple combination of the two, as both mechanisms have been well studied in previous works. Unfortunately, the revised version still does not meet this expectation. The reviewer sincerely suggests that the authors reconsider the benefits of combining these two mechanisms. The authors should ask whether the same results would emerge if the stiffening mechanism were replaced with another method. What are the exclusive advantages of integrating these two principles? The current combination seems too simple, as the performance appears to result from a straightforward combination of two previously published papers on ERF and ribbon actuators (e.g., Figs. 2 and 3). The data in Figs. 4 and 5 focus on the stiffening effect on the actuation, while Figs. 6–8 demonstrate the influences of stiffness on the ribbon actuators. In fact, the effects of stiffness on ribbon actuators have already been well studied in prior research, including the authors' own published manuscript:

Xu, Yuejun, Etienne Burdet, and Majid Taghavi. "Electromechanical model for electro-ribbon actuators." *International Journal of Mechanical Sciences* 275 (2024): 109340.

As a result, the current integration of ERF does not introduce a new feature compared to, for example, using wider or thicker ribbons in the ribbon actuators.

2. Additionally, Figs. 7 and 8 demonstrate a damping effect, but this is neither discussed nor modeled in the earlier sections. This phenomenon needs further exploration in the context of the demonstrations.

3. Do the results concerning stiffness in Fig. 2d align with those in Fig. 2b?

4. This work lacks model verification and fails to explain the large discrepancy between the modeled and experimental results.

5. Please ensure proper use of significant figures, such as in lines 219 and 220.

6. Minor Comments: Ensure format consistency throughout the paper. For example, there should be a space between numbers and units in several instances (e.g., lines 383, 404, and 405).

Reviewer #4

(Remarks to the Author)

Version 2:

Reviewer comments:

Reviewer #1

(Remarks to the Author)

The reviewer's concerns have been addressed by the authors. The reviewer recommends for publication.

Reviewer #3

(Remarks to the Author)

The reviewer has no more questions.

Response to reviewers

Dear Editor and Reviewers,

We thank you for thoroughly reviewing our manuscript and providing insightful comments. To address all the comments, we have conducted new experiments on dynamic response, durability, and muscle-like function, and completed modeling and analysis to characterize its dynamic behavior. We have also proofread the manuscript and made minor changes throughout. Below, we have provided detailed responses to each of your comments. The revisions made to the manuscript and supplementary information are highlighted in red.

Reviewer #1

RIC1

In this manuscript, Xu et al. developed an electrostatic actuator with independent stiffness controls. The design concept was introduced, the fabrication processes were described and the actuation and stiffness modulation performance were characterized extensively. Overall, this paper demonstrates very clear novelty and is very well written. The reviewer believes that this paper can be potentially publishable in this journal if the following comments can be addressed properly.

In this paper, the key concept of independent stiffness modulation (as indicated by the title of this paper) was kept mixing with the impedance (stiffness and damping) modulation (as mentioned in abstract, introduction and in some experimental results). This can be really confusing when going through this paper. By looking at the title, the reviewer felt that 'ok, this paper is about an actuator that can control its stiffness'. Then in abstract (line 26-27), the reviewer felt 'ok, it can also vary the damping ratios. Later in the results and demonstration parts, the reviewer finally realized that 'the ERF can vary the stiffness and damping simultaneously in this design, but the increase in damping is only a byproduct of the stiffness increase'. So, this sort of flow (as experienced by the reviewer) can potentially lower the impression of this paper, thereby

reducing its impacts. As a result, the authors should be very careful with what the 'money shot' of this paper is.

Thank you for pointing out the lack of consistency in the description of the paper's key concept. Our primary objective was to achieve independent stiffness modulation with a high-performance electrostatic actuator using a straightforward monolithic structure to extend its force capabilities. The intrinsic compliance of soft actuators, while offering a flexible deformation range, limits the achievable force output. Therefore, the seamless integration of variable stiffness primarily enables actuators to adjust their stiffness according to different operational scenarios (e.g., increasing stiffness enhances the robot's load-carrying capacity and stability, thereby preventing excessive deformation). Variable damping, on the other hand, plays a crucial role in dynamic tasks, such as vibration and shock absorption, or to improve stability. In many applications involving actuators with high inherent compliance, the impact of variable stiffness is often more significant, especially in relation to force output [13]. Therefore, we emphasised stiffness modulation in the title. However, our design naturally influences both stiffness and damping which is now investigated further in the revised manuscript. Although these changes are interconnected, here we demonstrate a method to distinguish stiffness and damping changes in specific scenarios. In summary, we have made the following revisions:

1. We have revised the terminology throughout the manuscript for consistency, removing the word "impedance," and clarified the confusing phrases related to independent control of stiffness and modulation. Please refer to lines 104-106 in the manuscript.
2. To study damping and stiffness modulation in more detail, we introduced an equivalent dynamic model and conducted dynamic tests, including free response and frequency response tests, to quantify these changes under different stiffening voltages. These tests demonstrate that both stiffness and damping vary with stimulation, a characteristic of the electrorheological fluid (ERF) that enhances the actuator's performance. Please refer to the new additions in the manuscript in lines 292-329 and the new Figure 5.
3. Based on our findings regarding the differing trends in damping and stiffness variation, we conducted additional demonstrations using both the Electro-Ribbon Actuator (ERA) and the Electro-Stiffened Ribbon Actuator (ESRA) within a parallel actuator system. These demonstrations, presented in the new Figure 6, and detailed in lines 330-359 in the

manuscripts, illustrate how these designs can achieve nearly independent increases in stiffness and damping under specific conditions. Our approach underscores the advantages of developing flexible and multifunctional actuation solutions, which enhance performance across a wide range of applications, such as variable stiffness semi-active vibration isolators for vibration control.

In addition to the above additions, we have now included a corresponding Methods section which can be found at the end of the manuscript, at lines 495-502, 514-519 and 557-613 with further details provided in the supplementary information in lines 161-215 and Supplementary figures 3, 4, 11 and 12.

RIC2

To better demonstrate the advantages and novelty of this ESRA design, please compare the key performance indices as well as power consumptions of the ERF adopted in this work as the variable stiffness mechanism and electrostatic force-based clutches and jamming mechanisms.

Thank you for this suggestion to highlight the advantages and novelty of ESRA. The integration of additional components with actuators, especially when dissimilar materials are used, typically affects the actuator's overall performance. However, here the integration was achieved with minimal changes by using a technology compatible with the actuator, which has extended its capabilities, enabling complex behaviour with modulated force and frequency. In our design, we used the Electro-Rheological Fluid (ERF) due to its seamless integration with ERA, enabling electric control, fast response and dynamic variable stiffness. Unlike electrostatic clutches, which typically require multiple layers to enhance stiffness variation, complicating fabrication and reducing the actuator's compactness, ERF offers a more streamlined solution, and operates at similar voltage as electro-ribbon actuators. Moreover, the stiffness characteristics of electrostatic clutches almost disappear in the fully sliding state. Using a vacuum-controlled jamming mechanism often requires extended times to increase stiffness and even longer to reduce it. Furthermore, it necessitates bulky air pressure resources and control units that are not efficient to integrate with the electrostatic driving unit as it negatively affects the overall performance of the system by adding extra weight and complexity. In contrast, our approach uses the voltage control

of four electrodes to achieve actuation and stiffness changes. In the future, integrating the power supply into the actuator could enable an untethered operation.

Our approach also leverages the dynamic variable stiffness property, rather than simply locking the actuator in a specific position, allowing stiffness changes to improve the actuator's dynamic response. To date, there has been limited research exploring this feature. For example, SJBAM [20] employs pneumatic driving and layer jamming; however, the necessary equipment for actuation and stiffness variation is bulky, and the response times are slow. We have summarised these in the Supplementary Table 1, where we compared the key characteristics of recently developed variable stiffness soft robots, including actuation mode, variable stiffness mechanism, response time, and power consumption. The relevant discussion has also been included in the manuscript in lines 411-422.

RIC3

The reviewer noticed that the experimental results are not compared with modelling results in this paper. Please include some modelling results next to the experimental data in the Characterization of ESRA section.

Thank you for bringing this issue to our attention. We have now compared the quasi-static modeling results of the isometric test under 0 kV and 7 kV actuation and included them in the **new panels g, h of Fig.4**. Detailed modeling information is provided in the **Supplementary Information in lines 155-159**.

RIC4

Based on Figure 6, both stiffness and damping increase with the stiffening of the ESRA. However, in many circumstances, it might be more desirable to independently control the stiffness and damping of an actuator. Would this be realized by using the ESRA?

Thank you for the insightful comment. In our current setup, the input control does not allow for fully independent adjustments of stiffness and damping. However, as discussed in Response to *RIC1*, we have observed distinct trends in stiffness and damping variations when changing the

stiffening voltage. This unique characteristic can be leveraged to develop an actuation system with nearly independent control over damping and stiffness. This capability is demonstrated using a one-dimensional agonist–antagonist muscle system (Fig. 6 in Response to *RIC1*), where the Electro-Stiffened Ribbon Actuator (ESRA) and the Electro-Ribbon Actuator (ERA) are arranged in parallel. This configuration enables control over amplitude and resonance frequency through different actuation and stiffening methods, highlighting the potential for independent modulation of stiffness and damping in specific scenarios.

To further understand the variable stiffness and damping mechanisms of the ESRA, we developed equivalent dynamic models for a single ESRA and the agonist–antagonist muscle system. They provide deeper insights into how these properties can be harnessed and controlled in future applications which is included in the Supplementary Information 161-215 and Supplementary figures 11 and 12.

RIC5

Fabrication section can be adjusted to the bottom of this manuscript or in the supplementary.

We have followed this suggestion and moved the fabrication section to the end of the manuscript, incorporating it into the Methods section. Additionally, the “Materials and Components” section and the figure of the fabrication process have been moved to the Supplementary Information.

RIC6

What is the material of the electrodes in Fig. 3?

The electrodes in the ESRA are made of steel strips with varying thicknesses (30 μm – 50 μm). For clarity, we have added this information to Supplementary Fig. 5a (originally Fig. 3).

RIC7

Figure 6 (i & ii), it would be more convenient to have the mass written in the plots

Thank you for this suggestion. We have added the weight value to **Fig. 7c** (originally Fig. 6).

R1C8

Supplementary Figure 4 A, the legends in the first plot are in wrong colors

Thank you for highlighting this mistake. We have revised the legends in **Supplementary Fig. 1** (originally Supplementary Fig. 4) and updated the color scheme.

Reviewer #2

R2C1

What are the noteworthy results?

--This paper introduces the "Electro-Stiffened Ribbon Actuator (ESRA)," a new actuator that enhances current "electro-ribbon actuators (ERA)" by adding a variable stiffness function. The authors redesigned the ERA's beam to include a sandwich structure with two electrodes and electrorheological fluid (ERF) in the middle. When voltage is applied between the two electrodes, the viscosity of the ERF increases, thereby stiffening the actuator. This design largely retains the original structure, form factor, and functions of conventional ERAs. Additionally, the stiffening function can operate independently of the actuation, providing greater flexibility. The authors also discovered that the stiffening function could enhance actuation strength, resulting in a higher payload capacity than conventional ERAs.

We thank the reviewer for highlighting the key aspects of our work and the innovations introduced with the Electro-Stiffened Ribbon Actuator (ESRA).

R2C2

Will the work be of significance to the field and related fields? How does it compare to the established literature? If the work is not original, please provide relevant references.

--Adding variable stiffness to soft actuators can be significant. Soft robots with variable stiffness have better force transmission, therefore improving power output. The variable stiffness sandwich-structured beam with two electrodes and an ERF infill was developed by Huilan Jing

et al., as cited by the author in the introduction. The author has made several modifications to adapt this design for the ESRA. For example, the author uses PVC tape to encapsulate the electrode and VHB tape to seal the ERF, which reduces the beams' thickness, and simplifies the manufacturing process. The authors have also chosen the steel electrode that better suits the ESRA.

We thank the reviewer for their insightful comments on the significance and originality of our work.

R2C3

Does the work support the conclusions and claims, or is additional evidence needed?

--The author claims that the newly developed ESRA is "compact", "high-contraction", and is a "multifunctional muscle-like actuation solution". This work provided clear figures including actuator dimensions and manufacture process to show that adding variable stiffness does not significantly increase the form factor, and the figures and the supplementary videos show that the compactness of the actuator. Experiment in figure 5 has shown the high-contraction claim. Demonstrations in figures 6 and 7 uses ESRA to simulate muscles and has shown how variable stiffness contributes to the system by comparing the stiffened actuator with the non-stiffened actuator. Overall, the work did support its claims through experiment results.

--The first paragraph of the Results section makes the claim that "ESRA allows for independent modulation of stiffness and damping properties," but the experiments do not seem to justify this claim. In the experiments, stiffness and damping seem to be coupled together, not independently controlled. Further experiments or explanation should be provided to justify this claim or it should be removed.

Thank you for this insightful comment. We have studied this in more depth and conducted new experiments, modelling and analysis. Our design naturally influences both stiffness and damping. Although these changes are interconnected, here we demonstrate a method to distinguish stiffness and damping changes in specific scenarios. In summary, we have made the following revisions:

1. We have revised the terminology throughout the manuscript for consistency, removing the word “impedance,” and clarified the confusing phrases related to independent control of stiffness and modulation. Please refer to lines 104-106 in the manuscript.
2. To study damping and stiffness modulation in more detail, we introduced an equivalent dynamic model and conducted dynamic tests, including free response and frequency response tests, to quantify these changes under different stiffening voltages. These tests demonstrate that both stiffness and damping vary with stimulation, a characteristic of the electrorheological fluid (ERF) that enhances the actuator’s performance. Please refer to the new additions in the manuscript in lines 292-329 and the new Figure 5 with further details provided in the Supplementary Information in 161-196 and Supplementary figures 3, 4 and 11.
3. Based on our findings regarding the differing trends in damping and stiffness variation, we conducted additional demonstrations using both the Electro-Ribbon Actuator (ERA) and the Electro-Stiffened Ribbon Actuator (ESRA) within a parallel actuator system. These demonstrations, presented in the new Figure 6, and detailed in lines 330-359 in the manuscripts, illustrate how these designs can achieve nearly independent increases in stiffness and damping under specific conditions. To further understand the variable stiffness and damping mechanisms, we developed equivalent dynamic model for the agonist–antagonist muscle system. They provide deeper insights into how these properties can be harnessed and controlled in future applications which is included in the Supplementary Information in lines 198-215 and Supplementary figures 12. Our approach underscores the advantages of developing flexible and multifunctional actuation solutions, which enhance performance across a wide range of applications, such as variable stiffness semi-active vibration isolators for vibration control.

R2C4

Are there any flaws in the data analysis, interpretation and conclusions? Do these prohibit publication or require revision?

--I have not found errors in the data analysis and interpretation. All the data explicitly supports the arguments in this work.

We thank the reviewer for their thorough review and feedback on the data analysis and interpretation of our manuscript.

R2C5

Is the methodology sound? Does the work meet the expected standards in your field?

--The characterization of the actuator is well-rounded. I'm interested in the durability of the proposed actuator. It might be helpful to other readers if the authors could hang a weight on the actuator and let it actuate for an extended period, and report the number of cycles the actuator lasts, and the actuation consistency throughout the process.

We have now conducted durability tests to assess the longevity and actuation consistency of the ESRA. The results of these durability tests are presented in Fig.5f&g and Supplementary Fig. 6. We tested the actuator for over 1500 cycles, and did not observe any noticeable changes in its actuation performance throughout the process. The detailed procedure for the durability test is included in Methods section at lines 504-512.

R2C6

Is there enough detail provided in the methods for the work to be reproduced?

--Yes, the author has provided enough detail that I am confident that I can reproduce the actuator manufacturing and experiments.

Thank you.

R2C7

The authors can improve the visual presentation of the data. Currently, the color scheme between each figure is not unified. A consistent color scheme across all figures would enhance readability.

Thank you for pointing out this visual presentation issue. To address this, we have revised all the figures in the manuscript to adopt a consistent color scheme.

R2C8

In the plots at the bottom of Fig. 6 it would be easier for the reader to visually compare the responses in the “actuation”, “no stiffening”, and “stiffening” conditions if they overlapped, rather than plotted sequentially in time.

To improve the visual comparison of responses in the “actuation,” “no stiffening,” and “stiffening” conditions in Fig 7, we have revised the plots at the bottom of Fig. 7c (originally Fig. 6) to overlay them.

R2C9

In Fig. 7b it is unclear what all of the dashed lines mean. Please clarify in the diagram or caption.

The caption for Fig. 8b (originally Fig.7b) has been updated to provide a detailed explanation of what each dashed line represents.

Reviewer #3

R3C1

“This work proposed a stiffness-tunable actuator based on electrostatic ribbon actuator and variable stiffness technique of electrorheological fluid.

Honestly speaking, the reviewer cannot see the innovation, significance and contribution of the combination of the two techniques (electrostatic actuator and variable stiffness). Particularly, several previous works have already published the idea of combining soft robots and electrorheological fluid, with the similar fabrication method, e.g., Jing, Huilan, et al. "Variable stiffness and fast-response soft structures based on electrorheological fluids." Journal of Materials Chemistry C 11.35 (2023): 11842-11850.”

We thank the reviewer for their feedback and for highlighting the need to better articulate the innovations and contributions of our work, and clarify how our research differs from previous

studies and offers unique advantages to be employed in a range of applications. We believe we have now addressed these issues, and also improved our manuscript by adding new experiments, modeling, analysis and clarifications.

While previous works, such as the study by Jing et al., have explored the capability of using electrorheological fluids (ERF) in variable stiffness structures, their focus was primarily on passive applications like shape locking, without offering actuation capabilities and studying its interconnected behavior. In contrast, our work presents a novel and seamless integration of the electrostatic ribbon actuator (ERA) with ERF-based variable stiffness in a new multifunctional actuation design, which achieves enhanced performance and control employable in a wide range of applications. We compared ESRA with recently developed variable stiffness soft actuators in key performance indices summarized in Supplementary Table 1, and included a comparison discussion in the manuscript on lines 411 to 422.

We also would like to clarify the novelty of this work and highlight the main new additions to the manuscript as follows:

1. **Integration of Electrostatic Actuation with ERF:** Our approach combines electrostatic ribbon actuators (ERA) with electrorheological fluids (ERF) for variable stiffness, offering a novel method that enhances the functionality of soft electric actuators. This combination leverages the strengths of both technologies, such as compact structure, fast response and power efficiency, providing a new dimension to actuation and control in soft robotics. The integration of multiple components with actuators, especially when dissimilar materials are used, typically affects the actuator's overall performance. However, here the integration was achieved with minimal changes by using a technology compatible with the actuator, which has extended its capabilities, enabling complex behaviour with modulated force and frequency.
2. **Independent Modulation of Stiffness and Actuation:** The ESRA allows for the independent control of both the actuation process and stiffness & damping modulation. This capability provides a high degree of flexibility and adaptability, enabling the actuator to respond effectively to different operational demands.

3. Improved Static and Dynamic Response: Through the integration of ERF, the ESRA achieves improved load-bearing capacity and increased force output under the same actuation voltage. This enhancement makes the actuator suitable for a broader range of applications where both large deformation and high output force are essential. Additionally, the actuator's ability to quickly adjust stiffness and damping allows it to perform effectively in dynamic tasks, such as absorbing vibrations and shocks, thereby improving its utility in applications like wearable robotics and adaptive systems. In revising the manuscript, we have further studied the dynamic response of the ESRA to highlight its novel features in modulating stiffness and damping and demonstrated how these characteristics can be independently controlled in specific scenarios to achieve muscle-like operation. please refer to lines 293-360 along with the two new figures (5 and 6) in the main manuscript, the associated methods in the Methods section, and lines 161-215 and new Supplementary Figures 3, 4, 11 and 12 in the supplementary information.
4. Optimized Fabrication Process: We have developed a straightforward fabrication method with minimal actuation thickness, which has now been tested for durability over 1500 cycles for both actuation and stiffening, with no deterioration in performance (fig 5 f and g). This approach maintains high actuation performance while increasing load capacity, addressing a key limitation of soft actuators, and enhances practical dynamic applicability.
5. Versatility in Applications: The unique combination of multiple features in the ESRA opens new possibilities for applications as shown in demonstrations, including wearable technologies and artificial muscles, where an adaptive and responsive actuation is needed, as well as active/semi-active vibration isolation.

We trust these innovations offer significant contributions to the field, which we hope are clarified with the additional characterization, analysis and demonstrations.

Reviewer #4

R4C1

We thank the reviewer for their time and effort in evaluating our manuscript.

Response to reviewers

Dear Editor and Reviewers,

We thank you for thoroughly reviewing our manuscript and providing insightful comments. In response, we have made substantial revision to the manuscript. We developed a new model considering a time-dependent electric field and conducted additional experiments for model validation. We performed new analyses, tests, and demonstrations focusing on stiffness and damping modulation, including a demonstration that highlights more distinct variations in these parameters for vibration control.

Furthermore, we have expanded our comparisons of stiffness and damping variations with state-of-the-art and conventional devices. We have enhanced the structure and clarity of the manuscript, conducted thorough proofreading, and implemented minor revisions throughout.

Below, we have provided detailed responses to each of your comments. The revisions made to the manuscript and supplementary information are highlighted in blue.

Reviewer #1:

RIC1

The reviewer appreciates the authors' efforts on revising this manuscript by adding dynamic characterizations of the device and adding comparisons of this design against the state-of-the-art. However, the reviewer believes that this paper, in its current form, lacks the substantial significance and impacts to be published in this journal compared with the two electrostatic actuator related papers very recently published in Nature Communications, where the first one being more closely related:

Wang, X., Wang, Y., Zhu, M. et al. 2-dimensional impact-damping electrostatic actuators with elastomer-enhanced auxetic structure. Nat Commun 15, 7333 (2024).

Buchner, T.J.K., Fukushima, T., Kazemipour, A. et al. Electrohydraulic musculoskeletal robotic leg for agile, adaptive, yet energy-efficient locomotion. Nat Commun 15, 7634 (2024).

We appreciate your insightful comments and interest in related research. Although the referenced studies do not directly address variable stiffness, we would like to clarify how our work contributes to existing knowledge and advances this field and relates to these previous studies. In summary, our technology integrates both actuation and variable stiffness concepts, but it does more than simply combine them to achieve both functions. Instead, it produces a monolithic component that requires less material than the sum of both technologies combined, and enables multiple novel functions in soft actuators, which have not been demonstrated in similar, uncomplicated structures.

Firstly, electrostatic actuators, particularly those employing flexural bending mechanisms, have always been limited by structure stiffness, which significantly influences performance indicators. For instance, electro-ribbon actuators [1] require modifications in electrode thickness to achieve either large stroke or force. Similarly, the study by Wang et al. [2] cited by the reviewer describes auxetic electrostatic actuators (AELAs) encountering analogous constraints; AELAs can generate greater force by replacing soft elastic bands with stiffer ones but at the cost of displacement reduction. Although certain actuators may excel in the force or displacement range, no universal solution currently exists that simultaneously satisfies both requirements without compromising one for the other. Our development aims to address this limitation by altering the material properties dynamically to enhance structural stiffness without diminishing displacement capabilities, diverging from conventional approaches that rely on mechanical contraction or shape locking to increase stiffness. We have now clarified and provided further details in the Introduction at lines 65-79.

Secondly, although Buchner et al.'s research, the second paper cited by the reviewer, primarily concentrates on the control of agonist-antagonistic HASEL actuators, our work also demonstrates significant advantages in similar applications through enhanced functional characteristics. For example, in Buchner's study, multiple bags were layered and integrated with elastic bands acting as tendons to increase displacement, achieving joint angles between 15 and 20 degrees. In our demo, our actuator not only achieves a substantial 70-degree angle using a single unit but also offers adjustable stiffness or damping properties independent on the produced force, without requiring two antagonist actuators. This capability of force independent variable impedance is particularly critical in bionic mechanical legs, providing the ability to absorb shocks and disturbances encountered, such as when navigating obstacles [4]. We acknowledged Buchner's contributions within the context of electrostatic actuator applications in our Introduction at line 67.

Finally, we have expanded our results significantly to highlight its novelty. i) we refined our theoretical framework on electrostatic force by developing a simplified electrical model that accounts for time-varying electrostatic force affected by charge relaxation [5] and integrating it with a coupling electromechanical model. This improved model enables us to accurately quantify the performance of materials used in electrostatic actuators, including BP commonly used in biodegradable HASEL [6] and PI used in electrostatic bellow muscles (EBM) [7]. Our model has been experimentally validated across various materials, marking the first time these critical material properties have been identified and highlighted in the context of large-deformation actuation. These findings are detailed in the Results section (at lines 156-176) and further elaborated in the Supplementary Information (line 67-301) and Supplementary Figs. 8 to 12, as well as in the Supplementary Table 3. ii) We have conducted additional analyses and provided demonstrations concerning the control of stiffness and damping. The specifics of these improvements will be outlined in our response to *RIC2*. iii) we have added a thorough

comparison between our technology and other existing technologies for stiffness and damping modulation, and conducted new experiments to quantify stiffness variations using an alternative solution (i.e. electrostatic clutch) with those of our proposed method here. Detailed information on these comparisons will be provided in our response to *RIC3*.

With these revisions, we have also enhanced the structure and clarity of the manuscript to better address the concerns regarding the novelty and significance of our study, especially in the revised Introduction and Discussion sections, all highlighted in blue, including advancement in Introduction (lines 98-104) and potential improvement in Discussion (lines 510-515).

- [1] Taghavi, M., Helps, T., & Rossiter, J. Electro-ribbon actuators and electro-origami robots. *Sci. Robot.* **3**, eaau9795 (2018).
- [2] Wang, X., Wang, Y., Zhu, M. et al. 2-dimensional impact-damping electrostatic actuators with elastomer-enhanced auxetic structure. *Nat. Commun.* **15**, 7333 (2024).
- [3] Buchner, T.J.K., Fukushima, T., Kazemipour, A. et al. Electrohydraulic musculoskeletal robotic leg for agile, adaptive, yet energy-efficient locomotion. *Nat. Commun.* **15**, 7634 (2024).
- [4] Vanderborght B, Albu-Schäffer A, Bicchi A, et al. Variable impedance actuators: A review. *Robot. Auton. Syst.* **61**, 1601-1614 (2013).
- [5] Sîrbu I D, Preninger D, Danninger D, et al. Electrostatic actuators with constant force at low power loss using matched dielectrics. *Nat. Electron.* **6**, 888-899 (2023).
- [6] Rumley E H, Preninger D, Shagan Shomron A, et al. Biodegradable electrohydraulic actuators for sustainable soft robots. *Sci. Adv.* **9**, eadf5551 (2023).
- [7] Sîrbu I D, Moretti G, Bortolotti G, et al. Electrostatic bellow muscle actuators and energy harvesters that stack up. *Sci. Robot.* **6**, eaz5796 (2021).

RIC2

As stated by the authors, the key novelty of this paper is focused on the stiffness and damping modulation capability of the electrostatic actuator. However, as is shown in Fig. 5(b) the damping of this device is increased to its peak value at a very low voltage, yet the stiffness of the device increases almost linearly against the voltage, which makes that the stiffness modulation of this devices is almost certainly accompanied by a high additional damping. As the reviewer pointed out in the last round of review, it is more desirable to have individual control of the two properties, especially to be able to publish in such a top tier journal. The authors are suggested to focus more on this aspect if they insist to work on the concept of ES actuator with stiffness and damping modulation capabilities.

We appreciate your insightful comment and valuable suggestion. We agree with your observations regarding the trends in stiffness and damping. However, to clarify, Fig. 5b illustrates a variation in the damping ratio where the focus is on the trend, whereas the absolute values, as detailed in Supplementary Fig. 3, are not significantly large and do not impact

contraction. Conversely, the applied stiffening voltage enhances the electrostatic force by reducing the zipping angle, which accelerates the dynamic contraction as shown in Fig. 4c.

We want to clarify that, as with human muscles and almost all conventional or soft actuators (e.g., actuators in Supplementary Table 1), using a single ESRA cannot independently modulate stiffness and damping. In our previous revision, we employed an agonist-antagonist configuration to augment stiffness through muscle-like co-contraction. Compared to other agonist-antagonist actuators that only increase stiffness, our approach achieves either an independent increase in stiffness, a simultaneous increase in both stiffness and damping, or an increase in damping while maintaining minimal stiffness. In this revision, we present variations in relative stiffness and damping, which might not have been clearly articulated previously. These results are now thoroughly presented in the Results section (lines 427-436) and illustrated in Fig. 8c (shown below).

We acknowledge that the agonist-antagonist configuration might shift the equilibrium position during one-dimensional operation when stiffness is modified through actuation. To further study its stiffness and damping modulation and present a more adaptable system capable of adjusting stiffness and damping more distinctly without restricting displacement, we leveraged characteristics of our actuators, which resemble a Voigt element composed of parallel damping and spring components. Similar to conventional devices that use two Voigt elements in series to control damping and stiffness, we connect two actuators in series to emulate the functions of nearly independent variable stiffness and damping. We show this in the frequency response tests. Independent adjustment of the stiffening voltages allowed us to increase both stiffness and damping, increase stiffness while simultaneously reducing damping, or alternatively, increase damping while only slightly affecting stiffness (See the new fig 8d). Since such functionality is rare in actuation technology, we have drawn comparisons with some non-actuation devices [8][9] to highlight our system's unique capabilities. This passive configuration provides a comparable range of adjustment and functionality for vibration control as other variable stiffness and damping devices, yet with significantly simplified construction and an unprecedentedly compact size, as shown in Supplementary Table 2 and Supplementary Movie 5.

It is important to note that in dynamic situations, stiffness and damping are inherently interrelated; therefore, devices that aim for independent control over these properties often cannot fully eliminate their mutual influence without very precise initial adjustments to the stiffness and damping of all components [8]. Typically, the adjustment function of these devices [9] is not different from that of the technology presented here, though we demonstrate this function seamlessly integrated into an actuation device. Moreover, our stacked configuration preserves the actuation functionality with static stiffness variation unaffected. This stacking approach is both straightforward and effective and is commonly employed in various electrostatic actuators like HASEL, EBM, and AELA to increase actuation displacement. It

allows us to achieve greater displacement while maintaining simplified control over stiffness and damping.

We believe the demonstrated functionality and adaptability of our actuators represent significant advancements in both actuation and variable stiffness technologies. We hope that the additional analysis, tests, and demonstrations included in this revision clarifies its significance. These developments are detailed in the Results section (lines 406-409, 438-455), illustrated in Fig. 8, and are further elaborated in the Supplementary Information (lines 359-378) and Supplementary Fig. 16. A comparative analysis is also available in Supplementary Table 2, along with a demonstration in Supplementary Movie 5. We also added more discussion in the Discussion section (lines 499-508).

[8] Liu, Y., Matsuhisa, H., & Utsuno, H. Semi-active vibration isolation system with variable stiffness and damping control. *J. Sound. Vib.* **313**, 16-28 (2008).

[9] Sun, S., Yang, J., Li, W., Deng, H., Du, H., & Alici, G. Development of a novel variable stiffness and damping magnetorheological fluid damper. *Smart Mater. Struct.* **24**, 085021 (2015).

RIC3

Related to the previous comment on the comparison of this design against the state-of-the-art, the authors are suggested to add more comparisons on the stiffness/damping variation ratios. It can be noted from this paper that the stiffness modulation ratio is rather low for this design, so the authors should make strong justifications on why this ER fluid is adopted, instead of using, say, electrostatic layer jamming. Electrostatic layer jamming or clutch has rather rapid response (5 ms) and two layers will be able to increase the stiffness by 4-fold, so it is by no means bulky. See the two papers below.

*Hinchet, R. and Shea, H., 2020. High force density textile electrostatic clutch. *Advanced Materials Technologies*, 5(4), p.1900895.*

*Chen, C., Fan, D., Ren, H. and Wang, H., 2023. Comprehensive Model of Laminar Jamming Variable Stiffness Driven by Electrostatic Adhesion. *IEEE/ASME Transactions on Mechatronics*.*

We appreciate your insightful feedback regarding various variable stiffness techniques. In response to this comment, we have extended our comparison, now including stiffness and damping changes with advanced soft actuators and additional variable damping actuators. Due to the nonlinear deformation observed, stiffness values cannot be straightforwardly calculated by dividing force by displacement [10]. Instead, we employ stiffness and damping values obtained under a constant load through dynamic testing. As previously mentioned, aside from the most recent actuator exploring variable impedance [11], most systems focus solely on either stiffness or damping. Our actuators match the performance of others in terms of stiffness and damping, yet they offer the additional benefits of being lightweight, having a fast response, and providing

an enhanced dynamic response. These updates are detailed in Supplementary Table 1, with further explanations added in the Discussion at lines 475-484.

For further analysis on alternative stiffening solutions, as suggested by the reviewer, we conducted preliminary tests to implement an electrostatic clutch in the seamless beam integration and studied its bending stiffness. We tested two sliding configurations: a standard beam and a clamped beam, the latter to prevent layer separation during significant motion. The electrodes and beam dimensions used in our experimental setup are identical to those in the ERF beam (Fig. 2d), and the insulator material is the same as in the literature [12]. Our study shows that the initial stiffness of the two-layer beam is greater than that of the ERF beam, while at 8kV, its final stiffness is significantly lower, resulting in a small overall stiffness variation. The introduction of a clamp further increases the initial stiffness, leading to an even smaller variation. According to Chen et al. [12] (the reviewer's referenced paper), the observed four-fold increase in stiffness in two-layer materials occurs only during the initial linear slide stage, equivalent to a bending deformation of 1% of the beam length. For example, a stiffness increase of four times is observed in a 50 mm beam until a deformation of 0.5 mm; beyond this point, stiffness rapidly decreases to less than 1.2 times at a deformation of 4 mm [12]. This behavior is unsuitable for our actuator, which typically operates at nearly 40% of its potential. A potential approach to increase stiffness variation is by stacking multiple layers; however, this method results in a bulkier system with higher initial stiffness. Additionally, we noted that after voltage removal, the electrostatic clutch exhibits long-term residual friction, leading to hysteresis and an asymmetric buckling phenomenon during rebound. We have detailed the experimental setup and displacement and bending stiffness comparisons in Supplementary Fig. 17, and added further explanations and comparative references in the Introduction (lines 79-82) and Discussion (lines 480-482).

The ERF employed in our setup integrates seamlessly with the electrostatic actuator in terms of configuration, voltage values, control units, and response speeds, while also achieving greater stiffness variation compared to similar technologies in a compact form factor. The controllable viscoelastic properties of the material emulate muscular characteristics, enhancing the output force and providing resistance to impacts and disturbances. To clarify our choice to use ERF, we have included justifications in the Introduction at lines 83-84.

[10] Wolf S, Grioli G, Eiberger O, et al. Variable stiffness actuators: Review on design and components. *IEEE/ASME Trans. Mechatron.* **21**, 2418-2430 (2015).

[11] Do, B. H., Choi, I. & Follmer, S. An all-soft variable impedance actuator enabled by embedded layer jamming. *IEEE/ASME Trans. Mechatron.* **27**, 5529-5540 (2022).

[12] Chen C, Fan D, Ren H, et al. Comprehensive Model of Laminar Jamming Variable Stiffness Driven by Electrostatic Adhesion. *IEEE/ASME Trans. Mechatron.* (2023).

Reviewer #2:

R2C1

I co-reviewed this manuscript with one of the reviewers who provided the listed reports. This is part of the Nature Communications initiative to facilitate training in peer review and to provide appropriate recognition for Early Career Researchers who co-review manuscripts. All of our previous comments have been sufficiently addressed by the authors.

We thank the reviewer for their time and effort in evaluating our manuscript.

Reviewer #3:

R3C1

This work proposes the integration of Electro-Rheological Fluids (ERF) and ribbon electrostatic actuators for multi-mode actuation, including active actuation and stiffening.

The reviewer expected a novel feature that could not be achieved by either one of the actuation mechanisms alone or through a simple combination of the two, as both mechanisms have been well studied in previous works. Unfortunately, the revised version still does not meet this expectation. The reviewer sincerely suggests that the authors reconsider the benefits of combining these two mechanisms. The authors should ask whether the same results would emerge if the stiffening mechanism were replaced with another method. What are the exclusive advantages of integrating these two principles? The current combination seems too simple, as the performance appears to result from a straightforward combination of two previously published papers on ERF and ribbon actuators (e.g., Figs. 2 and 3). The data in Figs. 4 and 5 focus on the stiffening effect on the actuation, while Figs. 6–8 demonstrate the influences of stiffness on the ribbon actuators. In fact, the effects of stiffness on ribbon actuators have already been well studied in prior research, including the authors' own published manuscript:

*Xu, Yuejun, Etienne Burdet, and Majid Taghavi. "Electromechanical model for electro-ribbon actuators." *International Journal of Mechanical Sciences* 275 (2024): 109340.*

As a result, the current integration of ERF does not introduce a new feature compared to, for example, using wider or thicker ribbons in the ribbon actuators.

We appreciate your insightful comments. We would like to further clarify how our study advances the field of electrostatic and variable stiffness actuators. In summary, our technology integrates both actuation and variable stiffness concepts, but it does more than simply combine them to achieve both functions. Instead, it produces a monolithic component that requires less material than the sum of both technologies combined, and enables multiple novel functions in soft actuators, which have not been demonstrated in similar, uncomplicated structures.

Firstly, although developments in flexural actuators, such as electro-ribbon actuators [1] and very recent AELAs [2], have showcased high performance, these are inherently limited by the dependence of performance metrics on actuator design. For instance, achieving high force or high stroke necessitates specific designs, such as using stiffer electrodes for ERA or stiffer elastomers for AELA, often at the expense of displacement. Consequently, while a specific actuator may excel in one performance metric, no universal or standard actuator can simultaneously meet multiple criteria effectively. To address this longstanding limitation, we have introduced an approach involving dynamically variable stiffness that can be adjusted without altering the equilibrium position, thus enhancing force capacity without compromising displacement. This novel capability is now highlighted in the Introduction at lines 65-79.

Secondly, we propose a seamless integration of ERF and electrostatic actuators in terms of configuration, voltage values, control units, and response speed. In comparison to similar technologies such as the electrostatic clutch, our system exhibits significantly greater stiffness variation, as demonstrated in the quantitative comparisons in the revised manuscript (Supplementary Fig. 17). Additionally, electrostatic clutches often struggle with large motions due to residual friction, which can lead to buckling and asymmetric deformation of the beam during rebound. Other variable stiffness mechanisms, like particle jamming and shape memory alloys, typically require bulky control units or have slow response times, which compromise system compactness and responsiveness. With the proposed seamless integration, our technology enhances both static and dynamic responses and offers controlled viscoelasticity akin to muscle tissue, increasing stiffness while improving resistance to impacts and disturbances. We have detailed these advantages and provided comparative analyses in the Introduction (lines 79-84) and the Discussion section (lines 475-484) as well as in Supplementary Fig. 17.

Third, we acknowledge your point that in static cases, changing the stiffness—akin to using a thicker electrode—alters the force output. However, we achieve this adaptation without the need to redesign the actuator for each use. Furthermore, as previously discussed, this represents a limitation of such actuators where dynamic changes in stiffness are crucial for achieving both high stroke and force. Traditional variable stiffness actuators typically require complex setups involving two motors connected to two nonlinear springs [3], equating to four units to function. In contrast, our actuator possesses nonlinear mechanical properties and independent variable stiffness, enabling it to perform the same functions with a single unit. Additionally, our actuator offers enhanced dynamic performance, facilitating a range of functions within a compact design. This capability is proven by new experiments and analysis, will be elaborated upon in our detailed response to comments regarding stiffness and damping below.

Furthermore, we have expanded our results significantly in the revised manuscript, which further highlights the significance of this study. i) we refined and validated our theoretical framework on the electrostatic force by developing a new electrical model that accounts for time-varying

electrostatic force affected by charge relaxation, and integrating it with a coupling electromechanical model. The specifics of these enhancements will be outlined in our response to *R3C4*. ii) We have conducted additional analyses and provided demonstrations concerning the control of stiffness and damping. The specifics of these enhancements will be outlined in our response to *R3C2*. iii) We have conducted a thorough comparison between our technology and existing technologies for stiffness and damping modulation and conducted new experiments to quantify stiffness variations in an alternative technology. Our findings demonstrate that the stiffness and damping variations of our system are comparable to state-of-the-art technologies, while simultaneously offering additional functionalities and enhancements. Detailed comparisons are provided in Supplementary Table 1, Supplementary Table 2 and Supplementary Fig. 17, with extensive explanations added in both the Introduction and Discussion sections.

With these revisions, we have also enhanced the structure and clarity of the manuscript to better address the concerns regarding the novelty and significance of our study, especially in the revised Introduction and Discussion sections, including advancement in Introduction, lines 98-104 and potential improvement in Discussion, lines 510-515.

- [1] Taghavi, M., Helps, T., & Rossiter, J. Electro-ribbon actuators and electro-origami robots. *Sci. Robot.* **3**, eaau9795 (2018).
- [2] Wang, X., Wang, Y., Zhu, M. et al. 2-dimensional impact-damping electrostatic actuators with elastomer-enhanced auxetic structure. *Nat. Commun.* **15**, 7333 (2024).
- [3] Wolf S, Grioli G, Eiberger O, et al. Variable stiffness actuators: Review on design and components. *IEEE/ASME Trans. Mechatron.* **21**, 2418-2430 (2015).

R3C2

Additionally, Figs. 7 and 8 demonstrate a damping effect, but this is neither discussed nor modeled in the earlier sections. This phenomenon needs further exploration in the context of the demonstrations.

We appreciate the opportunity to further refine our manuscript based on your feedback. To enhance clarity, we have included a description of damping early in the manuscript, specifically in the Introduction at lines 98-104. We have also restructured the Results section for better flow, reordering the original Figures 7 and 8 as Figures 6 and 7, respectively. These figures now more effectively illustrate the muscle-like functional characteristics and the enhanced anti-impact and anti-disturbance properties resulting from improved damping and stiffness. Additionally, what was originally Figure 6 has been renumbered as Figure 8. This figure now includes new experiments demonstrating the diversity of stiffness and damping regulation, further details of which are discussed below. Corresponding explanations have also been added to the Discussion section to align with these changes.

Due to the highly dynamic nature of damping and the charging effects within the dielectric, as revealed by our new electrical model, electrostatic forces vary over time (Supplementary Equations (7) and (9)), complicating theoretical predictions. This variation presents challenges in accurately modeling damping dynamically. As a result, we characterize damping experimentally. The outcomes of these experiments are detailed in Figure 5 and the equivalent analysis in Supplementary Information (lines 303-338) and Supplementary Figs. 3, 4 and 14. Additional new tests and demonstrations are explained below.

Figure 8 shows a wider modulation capability that can be achieved with multiple actuators. In our previous manuscript, we showed agonist-antagonist configurations to regulate different combinations of stiffness and damping, including damping regulation that is often not feasible with other agonist-antagonist structures. In the revised manuscript, Figure 8c more clearly illustrates these relative changes in stiffness and damping. Corresponding details are added in Result section at lines 427-436.

Furthermore, we have expanded the multi-actuator configuration to a more versatile setup by connecting two actuators in series with a heavy load, creating a freely movable system. This configuration allows for independent adjustment of the variable stiffening voltage for each actuator, enabling precise control over system stiffness and damping. For instance, we can increase stiffness while reducing damping, or enhance damping while keeping stiffness variation minimal, thus achieving variable stiffness and damping functionalities. In our demonstrations, we successfully achieved vibration amplification and suppression using these adjustable properties. We also conducted frequency response tests to measure the variation in damping and stiffness as a function of the stiffening voltages. Additionally, we compared the characteristics of this setup with those of existing conventional variable stiffness and damping vibration isolation devices. Our system exhibits similar rates of change in stiffness and damping, while maintaining actuation and static stiffness change capabilities, which significantly simplifies the overall structure. This feature, currently not available in other soft actuators, is crucial for applications akin to artificial muscles [4]. The new demonstrator is detailed in Figure 8 and Supplementary Movie 5. We provide a comprehensive description in the Results section (at lines 406-409, 438-455), and Supplementary Information (lines 359-378). Comparative data are included in Supplementary Table 2. Discussions correlating these findings are added to the Discussion section, line 499-508. We believe these enhancements better explain the effects of stiffness and damping modifications and the overall adaptability of our actuation system.

[4] Higuera-Ruiz D R, Nishikawa K, Feigenbaum H, et al. What is an artificial muscle? A comparison of soft actuators to biological muscles. *Bioinsp. Biomim.* **17**, 011001 (2021).

R3C3

Do the results concerning stiffness in Fig. 2d align with those in Fig. 2b?

Yes, the caption for Fig. 2b has been updated to provide a detailed explanation: The circles represent the stiffness of the three-point bending shown in Fig. 2d.

R3C4

This work lacks model verification and fails to explain the large discrepancy between the modeled and experimental results.

We thank the reviewer for the valuable feedback. Our previous modeling was based on the electromechanical model of the electro-ribbon actuators. In this revision, we refined our theoretical framework on electrostatic force by developing a simplified electrical model that accounts for time-varying electrostatic force affected by charge relaxation [5] and integrating it with the coupling electromechanical model. This enhancement enables us to accurately quantify the performance of materials used in other electrostatic actuators [6][7]. Our model has been experimentally validated across various materials. Our research is the first to identify and highlight these crucial material properties in large deformation actuation. These findings are detailed in the Results section (lines 156-165, 170-176), and further elaborated in the Supplementary Information (lines 67-301) and Supplementary Figs. 8 to 12, as well as Supplementary Table 3.

In our model, we initially assumed uniform bending stiffness, which may not always hold under experimental conditions and could lead to discrepancies between our simulated and experimental data. Additionally, edge effects in the electric field may also influence the results. Despite these factors, the discrepancies observed are within a reasonable range, suggesting that our enhanced model remains robust. To address and clarify these points, we have expanded the discussion in the Supplementary Information (lines 289-291, 297-301).

[5] Sîrbu I D, Preninger D, Danninger D, et al. Electrostatic actuators with constant force at low power loss using matched dielectrics. *Nat. Electron.* **6**, 888-899 (2023).

[6] Rumley E H, Preninger D, Shagan Shomron A, et al. Biodegradable electrohydraulic actuators for sustainable soft robots. *Sci. Adv.* **9**, eadf5551 (2023).

[7] Sîrbu I D, Moretti G, Bortolotti G, et al. Electrostatic bellow muscle actuators and energy harvesters that stack up. *Sci. Robot.* **6**, eaz5796 (2021).

R3C5

Please ensure proper use of significant figures, such as in lines 219 and 220.

Thank you for your suggestions. To improve clarity, we have included more detailed references to the specific sub-figures in the manuscript.

R3C6

Minor Comments: Ensure format consistency throughout the paper. For example, there should be a space between numbers and units in several instances (e.g., lines 383, 404, and 405).

Thank you for your careful reading and pointing out this issue. We have revised the manuscript to ensure format consistency.

Reviewer #4:

R4C1

We thank the reviewer for their time and effort in evaluating our manuscript.